# Migratory dendritic cells acquire and present lymphatic endothelial cell-archived antigens during lymph node contraction

Ross M. Kedl[1], Robin S. Lindsay[1,2], Jeffrey M. Finlon[3], Erin D. Lucas [1,3], Rachel S. Friedman[1,2] & Beth A. Jirón Tamburini[1,3]

Antigens derived from viral infection or vaccination can persist within a host for many weeks after resolution of the infection or vaccine responses. We previously identified lymphatic endothelial cells (LEC) as the repository for this antigen archival, yet LECs are unable to present their archived antigens to CD8[+] T cells, and instead transfer their antigens to CD11c[+] antigen-presenting cells (APC). Here we show that the exchange of archived antigens between LECs and APCs is mediated by migratory dendritic cells (DC). After vaccination, both migratory basic leucine zipper ATF-like transcription factor 3 (BatF3)-dependent and BatF3-independent DCs are responsible for antigen exchange and cross-presentation. However, exchange of archived viral antigens is mediated only by BatF3-dependent migratory DCs potentially acquiring apoptotic LECs. In conclusion, LEC-archived antigens are exchanged with migratory DCs, both directly and through LEC apoptosis, to cross-present archived antigens to circulating T cells.

[1] Department of Immunology and Microbiology, University of Colorado Anschutz Medical Campus, School of Medicine, 12800 E. 19th Ave, Aurora, CO 80045, USA. [2] Department of Biomedical Research, National Jewish Health, 1400 Jackson Street, Denver, CO 80206, USA. [3] Department of Medicine, Division of Gastroenterology and Hepatology, University of Colorado Anschutz Medical Campus, School of Medicine, 12700 E. 19th Ave., Aurora, CO 80045, USA. Robin S. Lindsay and Jeffrey M. Finlon contributed equally to this work. Correspondence and requests for materials should be addressed to R.M.K. (email: ross.kedl@ucdenver.edu) or to B.A.J.T. (email: beth.tamburini@ucdenver.edu)

**D**uring the initiation of an immune response against viral challenge, numerous factors contribute to the swelling of local secondary lymphoid tissues and the resident stromal cells must expand to accommodate the influx of cells[1–3]. Production of vascular endothelial growth factors by migrating mononuclear cells and infiltrating B cells results in the growth of lymphatic vessels and blood vessels[1,2].

The recruitment of dendritic cells (DC) to the lymph node (LN) during an active immune response results in engagement of podoplanin (PDPN) on lymphatic endothelial cells (LEC) and fibroblastic reticular cells (FRC), causing relaxation of the FRC network, stromal cell division, and LN swelling[4–6]. However, the contraction of the stromal network is still not well understood. Even less clear is the effect of this process on the contracting

**Fig. 1** Archived antigen acquisition and DC antigen presentation. **a** Representative image of raw data from wild-type mice challenged with antigen (10 μg/site) alone, antigen with poly(I:C) (2 μg/site), and anti-CD40 (2 μg/site) or antigen (10 μg/site) with vaccinia virus (1 × 10⁴ CFU). One week after challenge, mice were harvested and flow cytometry was performed to visualize antigen acquisition. Shown are cells gated as in Supplementary Fig. 1B (CD45⁻, PDPN⁺, CD31⁺). **b** Number of antigen-positive lymphatic endothelial cells per lymph node was calculated from each challenge in **a**. **c** Mice were immunized weekly as indicated with antigen and poly(I:C)/aCD40 (as above). LECs were gated as CD45⁻, CD31⁺, PDPN⁺, blood endothelial cells (BECs) were gated as CD45⁻, CD31⁺, PDPN⁻, FRC/marginal reticular cells (MRC)/FDCs were gated as CD45⁻, CD31⁻, PDPN⁺, macrophages (Mac) were gated as CD11c⁻, B220⁻, CD11b^hi, F4/80⁺, DCs were gated as CD11c^hi, B220⁻, F4/80⁻, B cells were gated as B220⁺, and T cells were gated as CD3⁺ cells (Supplementary Fig. 1). The percent of antigen-positive of each cell type was calculated as a percent of the total cell type. Error bars shown in figure are mean ± standard deviation from two independent experiments with three mice per group per time point. **d** Schematic of the experimental design. **e** Wild-type mice were irradiated and K^bm8 (cannot effectively present the dominant ovalbumin peptide) bone marrow was used to reconstitute the hematopoietic compartment. OT1 T cells were transferred into these mice 2 weeks after immunization with ovalbumin (10 μg/site), poly(I:C)/anti-CD40 (2 μg/site), and 1 day later WT BMDCs were transferred subcutaneously (1 × 10⁶ per site). OT1 division of unimmunized mice that received BMDCs is shown in black, mice immunized, but did not receive BMDCs is shown in light gray, and mice immunized and received BMDCs is shown in dark gray. **f** Quantification of **e**. Shown in **e** is a representative experiment and in **f** quantification is shown from two independent experiments with three chimeric mice per group, where error bars are mean ± standard error of the mean. Statistical analysis was done using an unpaired t-test

lymphocyte population and their formation of productive and protective immune memory.

LN stromal cells produce and capture various chemokines. Specifically, follicular DCs (FDC) within the secondary follicle secrete chemokine (C-X-C motif) ligand 13 (CXCL13), attracting activated CXCR5[+] B and T cells into the secondary follicle to initiate the complex process of class switch recombination and somatic hypermutation[7,8]. Fibroblastic reticular cells secrete chemokine (C-C motif) ligand 19 and 21 (CCL19/21) and interleukin 7 for recruitment of CCR7[+] cells[9–12]. Lymphatic endothelial cells (LECs) in the cortical sinus of the LN produce sphingosine-1-phosphate (S1P), resulting in naive T cells, or activated T cells that have lost CD69 expression, to exit the LN and reenter the circulation[13]. LECs also produce chemokines such as CCL21[14], CXCL12[15], and CCL1[16] to influence DC recruitment to the LN. Functionally, LECs can present endogenous antigens and induce tolerance in both autoreactive T cells presented with peripheral tissue antigens[17–20] and tumor-specific T cells[21,22]. LECs are also reported to present exogenously derived antigens to CD8[+] T cells, though varying results have been seen depending on the experimental model used[21–23]. We previously demonstrated a function for LECs during the course of an immune response, a function for which we coined the term "antigen archiving"[24].

During the process of LN expansion and inflation, LECs capture and retain viral and vaccine-associated antigens for weeks after the resolution of the adaptive immune response. The long-term persistence of viral-associated antigens had long been known, but was a function largely ascribed to FDCs[25–30]. By contrast, we showed that persisting viral and subunit vaccine-related antigens are captured and stored, or "archived", by LECs for extended periods of time[24]. We also showed that archived antigen-bearing LECs are not capable of antigen presentation to CD8[+] T cells, but rather negotiate antigen exchange with CD11c[+] antigen-presenting cells (APC), which could cross-present antigens[17,23,24]. This idea is not without precedent, as antigen exchange between LECs and DCs for peripheral tissue antigens has been shown to be required for inducing CD4[+] T-cell anergy[20]. However, in our model, LEC-DC exchange of foreign antigens results in the stimulation of circulating memory CD8[+] T cells, augmenting protective immunity during the window of archived antigen persistence in the host[24]. These studies revealed a previously undocumented function for LECs that influences the maintenance of protective immunity. What remains unclear is both the subset of the CD11c[+] APC involved in antigen exchange with archived antigen-bearing LECs, as well as the mechanism by which antigens are removed from the LEC and received by the APC for presentation to CD8[+] T cells.

CD11c[+] DC subsets can be split into three major groups; conventional DC 1 (cDC1), conventional DC2, (cDC2) and plasmacytoid DC (pDC)[31]. Although antigen presentation by pDCs to T cells has been documented, the major known function is type I interferon (IFN) production during viral infection[32]. cDC1 development is mediated by the transcription factors interferon regulatory factor 8 (IRF8) and basic leucine zipper ATF-like transcription factor 3 (BatF3), and the lineages include both LN-resident and migratory subsets[31,33]. Aside from the canonical marker expressed by all mouse DC subsets (CD11c), both resident and migratory cDC1 cells express X-C motif chemokine receptor 1 (XCR1) and are either CD8[+] (LN resident) or CD103[+] (migratory)[33,34]. cDC2 development and function is regulated by IRF4, and resident and migratory cDC2 subsets are marked by the expression of signal regulator protein alpha (SIRPα) and CD11b[33,35–37]. The most probable subset involved in antigen exchange with LECs is the cDC1 subset. cDC1s are highly efficient in acquisition and processing of exogenous

antigens into class I major histocompatability complex (MHC), a process known as cross-presentation[38]. cDC1s are also unique in their capacity to acquire cell-associated antigens within either lymphoid[39,40] or peripheral[41] tissues. However, cDC2s may also mediate LEC-antigen exchange, as migratory CD11b[+] DCs can cross-present soluble antigens[42] and LN-resident cDC2s locate primarily to the subcapsular sinus of the LN[43,44], the same region as the LECs bearing archived antigens[24].

Here we show presentation of archived antigens to circulating T cells as a result of antigen exchange between LECs and DCs. Analyses of Ccr7[−/−] and BatF3[−/−] mice show that migratory DCs acquire vaccine antigens from LECs independent of BatF3 function. However, exchange of virus-derived antigens between DCs and LECs is mediated only by migratory BatF3-dependent DCs, suggesting possible differences in the mechanism of antigen exchange between vaccination and virus challenge. Finally, LEC-DC antigen exchange is, at least in part, mediated by the apoptotic cell receptor, C-type lectin domain containing 9A (Clec9a), expressed predominantly on BatF3-dependent DCs. Our data thus identify mechanisms by which the lymphatic stroma and hematopoietic cells interact to influence adaptive immunity.

## Results

**LEC expansion and antigen archiving.** The rapid expansion of antigen-specific lymphocytes following vaccination or viral infection is accompanied by the expansion of multiple stromal cell subsets, including LECs[1–3,24]. Following the resolution of the adaptive response and concomitant contraction of lymphocytes, the size and cellularity of the secondary lymphoid tissues in mice decreases with increasing time after viral challenge or vaccination. Over the course of weeks post immunization/viral challenge, vaccine/viral-related antigens are captured and maintained in association with LEC, a process we have termed antigen archiving. This can most readily be observed using fluorescently labeled antigens in conjunction with either a combination adjuvant (poly (I:C)/anti-CD40), or with vaccinia virus (VV) challenge. Using flow cytometry to focus specifically on the LECs (Fig. 1; Supplementary Fig. 1), their acquisition of fluorescent antigens can readily be observed (Fig. 1a, b). Consistent with our previous observations[24], antigens are only archived on LECs, and do not persist in whole protein form on any other stromal or hematopoietic cell type beyond the first week after immunization (Fig. 1c; Supplementary Fig. 1). However, as our data below demonstrate, APC acquire antigens at later time points for processing and presentation, resulting in loss of fluorescence.

Despite the fact that LECs are the only cell type within the lymphoid tissue to possess archived antigens, they do not process and present the exogenous antigens in the context of class I MHC, a function conventionally known as cross-presentation. This is observed through the use of mice expressing a mutant form of the class I K[b] molecule, K[bm8]. APCs derived from the K[bm8] background are incapable of effectively presenting the dominant peptide derived from ovalbumin (SIINFEKL) to the OT1 T-cell receptor (TCR) transgenic T cells[45]. By reconstituting a lethally irradiated WT C57BL/6 host with K[bm8] bone marrow, we create a host in which only the stromal cells express the wild-type (WT) Kb class I capable of SIINFEKL presentation and OT1 stimulation, while all hematopoietic APCs are K[bm8] and cannot elicit OT1 division (Fig. 1d–f). Consistent with our recent publication[24], immunization of WT:K[bm8] bone marrow chimeras resulted in the archiving of ova antigens on the LECs (Supplementary Fig. 2). The archived antigens on the WT:K[bm8] bone marrow chimaera was not cross-presented to T cells as evidenced by the complete lack of proliferation of transferred OT1s (Fig. 1e, f; Supplementary Fig. 3). However, subcutaneously

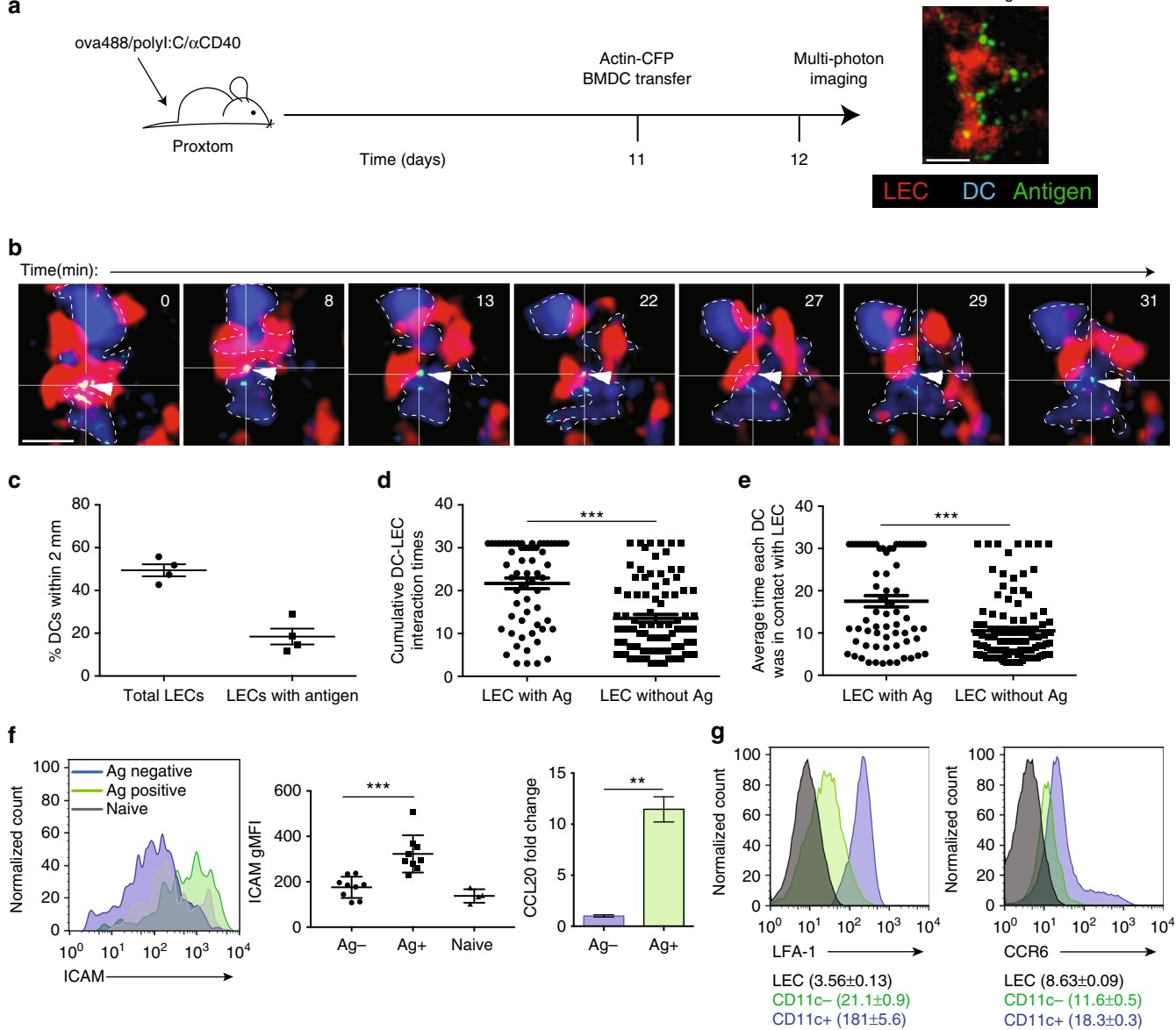

**Fig. 2** DCs interact with antigen-bearing LECs. **a** schematic of experimental design and antigen-bearing LEC, where LEC is red and antigen is green. Scale bar on image is 10 μm. **b** Still images from movie demonstrating prolonged association of DC-blue with white dashed outline, with antigen-green with white triangle and LEC-red. Scale bar on images is 15 μm. See Supplemental Movies for other examples. **c** Graph demonstrating the percentage of DCs within 2 μm of total LECs or only LECs that have antigen. Each dot represents the average from an individual lymph node. **d** Quantitation of DC–LEC cumulative interaction times from **c**. Analysis evaluated the duration that each DC interacted with each LECs that did or did not have antigen. **e** Quantitation of DC–LEC average interaction time. Analysis evaluated the average length of each interaction between DC and LEC. Seven-time lapse movies from four lymph nodes were imaged and quantified using IMARIS and Matlab software. **f** ICAM1 and CCL20 expression by antigen+ LECs compared to antigen—LECs 2 weeks after immunization. **g** Expression of C-C chemokine receptor (CCR6) and lymphocyte function-associated antigen–1 (LFA-1) on bone marrow-derived DCs grown for 6 days in GM-CSF. Statistical analysis was done using an unpaired t-test with a p-value of <0.0001. Error bars represent mean ± standard error of the mean

injected WT C57BL/6 bone marrow-derived DCs (BMDC) can traffic to the draining LN, acquire LEC-associated ovalbumin, and cross-present it to T cells (Fig. 1d–f).

**Visualization of DC–LEC interactions and antigen exchange.**
The success of the BMDCs transferred into the WT:K$^{bm8}$ bone marrow chimera to capture LEC-archived antigens presented an ideal model system in which we could utilize multi-photon microscopy to visualize this exchange. We injected Alexafluor 488-labeled ovalbumin (ova)+adjuvant into the footpad of the

proxtom mouse[46], a host that expresses tdTomato under the prospero homeobox protein 1 (prox1) promotor (a transcription factor that distinguishes LEC from other cells in the LN). Eleven days later, we injected BMDCs derived from the actin-cyan fluorescent protein (CFP) mouse into the footpad. The following day we performed multi-photon microscopy on the extracted popliteal LN mounted on a coverslip (Fig. 2a). BMDCs were seen to co-localize with both antigen-bearing (Fig. 2b, c) and non-antigen-bearing LECs (Fig. 2c). However, the BMDCs interacted with antigen-bearing LECs for significantly (p < 0.0001, by unpaired t-test) longer periods of time compared to the non-

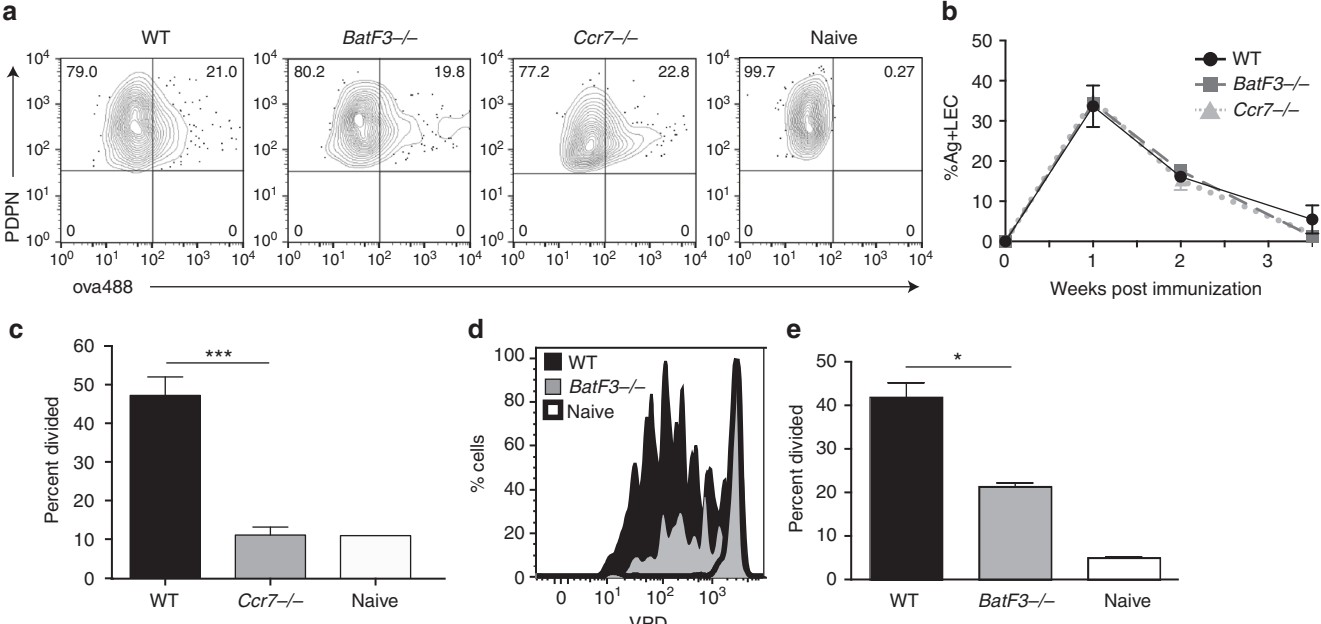

**Fig. 3** Archived antigen presentation is dependent on Ccr7. **a** Mice were immunized with ovalbumin conjugated to Alexafluor 488 (ova488)/poly(I:C)/αCD40 as in Fig. 1. Two weeks after immunization mice were killed, lymph nodes digested and stained for stromal markers. Shown are antigen-positive lymphatic endothelial cells from wild type, *BatF3⁻/⁻*, *Ccr7⁻/⁻*, and unimmunized (naive) mice. **b** Mice were immunized as in **a** and killed 1, 2, or 3.5 weeks post immunization to evaluate differences in percentage of antigen on lymphatic endothelial cells in different strains. There were no statistically different points between strains. **c** OT1 T cells were transferred into wild-type (black filled) or *Ccr7⁻/⁻* (gray) mice immunized 2.5 weeks prior and the percent divided was calculated using three mice per group and repeated three independent times. Statistical analysis was done using an unpaired *t*-test where *p* = 0.0003 between wild-type and *Ccr7⁻/⁻* mice. **d** Wild-type or *BatF3⁻/⁻* mice were immunized 3 weeks prior to killing and OT1 division was evaluated by violet proliferation dye (VPD) dilution. **e** Quantitation of **d**. Representative experiments are shown in **a** and **d**, and quantification of four independent experiments with 3–5 mice per genotype is shown. Statistical analysis was done using an unpaired *t*-test, where *p* = 0.0016 in **e**. Error bars shown in the figure are mean and standard error of the mean

antigen-bearing LECs (Fig. 2d, e; Supplementary Movie 1). This increase in interaction times between BMDCs and antigen-bearing LECs was observed whether measuring the cumulative interaction time during the 30 min course of the video captured (Fig. 2d) or when evaluating the average length of each interaction between LEC:BMDC (~20 min compared to ~10 min) (Fig. 2e; Supplementary Movies 1, 2). In many cases, we could even observe BMDCs interacting with antigens (arrow) on the LEC (Fig. 2b; Supplementary Fig. 4; Supplementary Movie 3). Further, when we transferred labeled, antigen-specific naive OT1 T cells, we were unable to locate them within the sinus or near LECs, but instead found them within the T-cell region where they were seen to interact with migrating BMDCs (Supplementary Movie 4). Interestingly, we found increased Chemokine CCL20 and intercellular adhesion molecule 1 (ICAM1) expression on the antigen-bearing LECs (Fig. 2f) and increased expression of both CCR6 and lymphocyte function-associated antigen 1 (LFA-1) on transferred BMDCs (Fig. 2g). This was not specific to ovalbumin conjugated to Alexafluor 488 as another antigen conjugated to a different fluorofore produced similar results (Supplementary Fig. 5). These data suggest a function for cell adhesion molecules and chemokines in regulating increased LEC–DC interaction times. We concluded from these data that migratory DCs interact preferentially with antigen-bearing LECs, an interaction that facilitates antigen exchange and eventual presentation to circulating T cells within the T-cell region of the node.

**Migratory DCs present vaccine-derived LEC-archived antigen.** As above, there are numerous subsets of DCs within the LN that might be expected to have the capacity to interact, and exchange

antigens, with LECs. Our observation with BMDCs suggested that APCs migrating past antigen-bearing LECs were able to interact with LECs, but did not identify which subsets of DCs were actually responsible for antigen exchange in vivo. To distinguish between migratory and LN-resident DC subsets involved in LEC antigen exchange, we utilized mice deficient in Ccr7. CCR7 is the chemokine receptor responsible for the recruitment of migratory DCs from the tissues and into the T-cell regions of the LN. As a result, the skin draining LN in *Ccr7⁻/⁻* mice are devoid of all migratory DC subsets (MHC class II high, CD11c⁺)[47]. We also utilized the BatF3 (*BatF3⁻/⁻*) host that is deficient in XCR1⁺ LN-resident (CD8⁺) and migratory (CD103⁺) subsets, but still possess XCR1⁻ CD11b⁺ resident and migratory DCs. Importantly, both BatF3-dependent DC subsets are highly efficient in facilitating cross-presentation of cell-associated antigens[39–41], whereas the CD11b⁺ DCs can cross-present soluble antigens[48]. WT, *Ccr7⁻/⁻*, and *BatF3⁻/⁻* mice were immunized with alexa-488-labeled antigens + adjuvant. The duration of antigen archiving on LECs was evaluated using flow cytometry over the duration of 3–4 weeks. Antigen exchange (at 2.5 weeks) was analyzed by OT1 division.

Antigen retention (Fig. 3a, b) occurred to the same degree in all strains, indicating that loss of DC subsets within the host did not adversely affect antigen archiving by LECs. However, there was a significant (*p* = 0.0003, by unpaired *t*-test) defect in the *Ccr7⁻/⁻* host in antigen exchange as observed by lack of T-cell division. The degree of antigen presentation in the *Ccr7⁻/⁻* host was no greater than a naive host (Fig. 3c). In contrast, the *BatF3⁻/⁻* mice, while substantially reduced from the WT, retained a reasonable amount of residual antigen presentation (Fig. 3d, e). We concluded from these data that the loss of antigen presentation

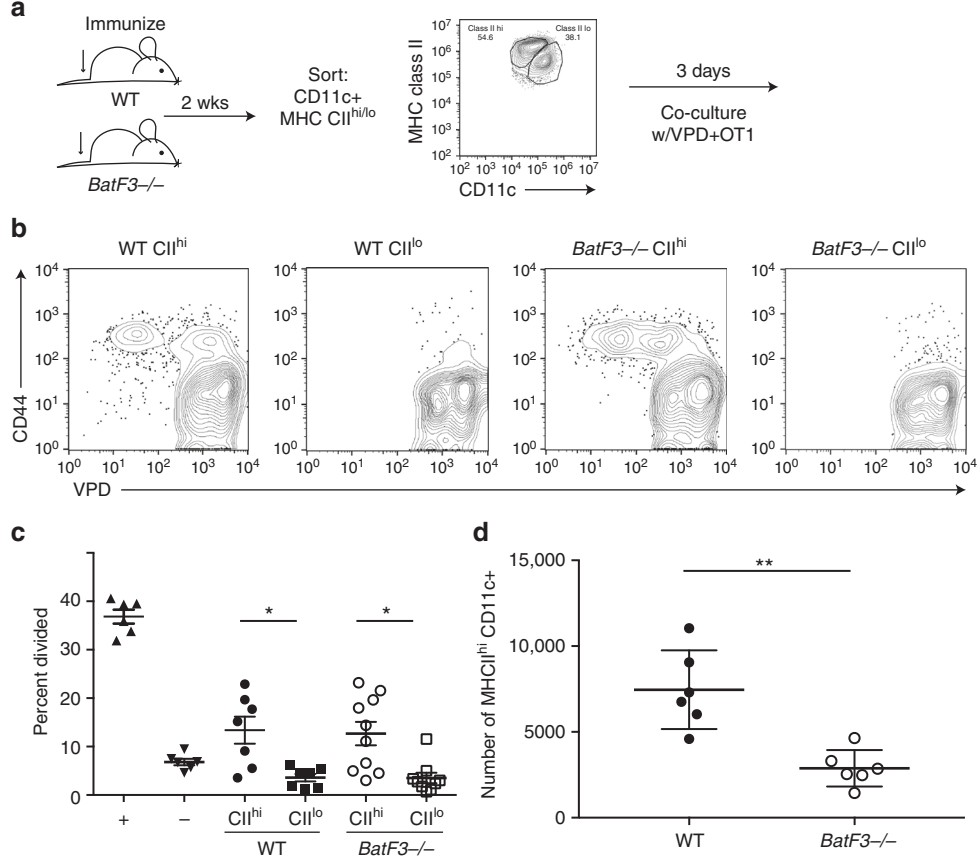

**Fig. 4** MHC II$^{hi}$ migratory DCs are required for archived antigen presentation. **a** Schematic representation of experimental design. WT of *BatF3*$^{-/-}$ mice were immunized subcutaneously as above 2 weeks prior to killing. Then lymph nodes were digested, T and B cells depleted, and then sorted on CD11c+ MHC class II high or intermediate expressing DCs. Approximately 20,000 DCs were co-cultured with 50,000 violet proliferation dye (VPD)-labeled OT1s for 3 days. **b** Flow cytometry of cells from **a**. Cells were gated on CD8$^+$, CD45.1$^+$ to evaluate activation (CD44) and division (VPD). **c** Quantification of samples from B where + is enriched lymph node DCs from a mouse immunized 1day prior and − is splenocytes from the *BatF3*$^{-/-}$ mouse. Experiment was repeated twice. Six mice per group were killed and DCs sorted based on the gating strategy in **a**. Each point represents one well with 20,000 DCs and 50,000 VPD-labeled cells (based on the number of DCs recovered). Statistical analysis was done using an unpaired *t*-test where $p = 0.0056$ for WT and $p = 0.0040$ for *Batf3*$^{-/-}$. **d** Lymph node cells from mice above were stained for flow cytometry to evaluate the frequency of DCs that were MHC class II high between groups and numbers were calculated. An unpaired *t*-test was used to calculate the *p*-value of <0.0001

in the *Ccr7*$^{-/-}$ host indicated a requirement for migratory APCs in the exchange and presentation of LEC-archived antigens derived from subunit vaccination. Additionally, the partial loss of antigen presentation in the *BatF3*$^{-/-}$ host supports this conclusion and further suggests that antigen exchange and presentation can be accomplished by both BatF3-dependent and BatF3-independent migratory DC subsets.

**Migratory DCs acquire and cross-present LEC-archived antigens.** The data above identified a requirement for migratory DCs in the process of antigen exchange with LECs. However, it is feasible that these migratory DC subsets are necessary to facilitate transfer of antigens from the LECs to another LN-resident DC. To address the question more directly, we isolated migratory and resident DCs, from WT and *BatF3*$^{-/-}$ mice, 2 weeks after subcutaneous vaccination with ovalbumin + adjuvant (Fig. 4a). After collagenase digestion and negative selection for all CD11c$^+$ cells, migratory (CD11c$^+$MHC class II$^{hi}$) and resident (CD11c$^+$MHC class II$^{lo}$) DCs were purified by flow sorting and equal numbers of each DC subset (from each host) were incubated with violet proliferation dye (VPD)-labeled OT1 T cells. Three days later, division of the OT1s was measured to indicate whether the isolated DC was cross-presenting in vivo-acquired antigens. Consistent with the in vivo results, migratory DCs from both WT

and *BatF3*$^{-/-}$ hosts initiated robust OT1 division, indicating their capacity to cross-present in vivo-acquired antigens (Fig. 4b, c). However, resident DCs from either host did not result in OT1 proliferation, indicating the lack of ovalbumin acquisition in vivo. As expected, the *BatF3*$^{-/-}$ had fewer total MHC class II$^{hi}$ migrating DCs compared to the WT (Fig. 4d), due to loss of all CD103$^+$ DCs. However, co-culturing the same number of DCs from WT and *BatF3*$^{-/-}$ hosts revealed no difference between the cross-presenting capacity of migratory DCs from the WT and *BatF3*$^{-/-}$ on a per cell basis (Fig. 4c). These data constitute direct evidence for the capture and cross-presentation of vaccine-derived LEC-archived antigens by all migratory DC subsets, both BatF3-dependent and BatF3-independent. It is worth noting as well that these data provide direct evidence that the proliferation of OT1s, when transferred into a host with LEC-archived antigens, is indeed an effective proxy for DC:LEC antigen exchange and cross-presentation.

**Migratory BatF3-dependent DCs present viral-archived antigens.** LEC antigen archiving occurs in response to either vaccination or viral challenge[24]. We have examined how subunit vaccine-derived soluble antigens are exchanged. We next asked if these principles were the same for viral infection and viral antigen exchange. Infection with an ovalbumin expressing strain of VV

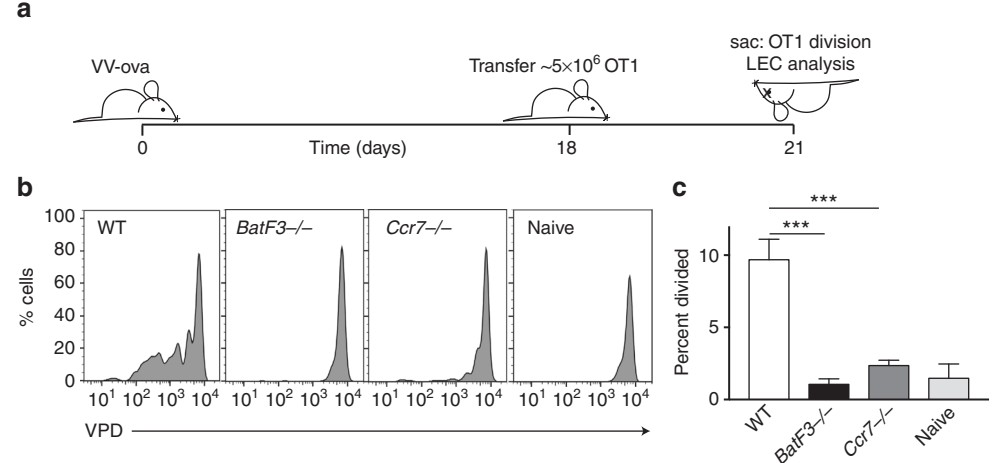

**Fig. 5** Antigen exchange following vaccinia infection requires both BatF3 and Ccr7. **a** Schematic of experimental design. **b** WT, $BatF3^{-/-}$, and $Ccr7^{-/-}$ mice were infected with vaccinia-ova and 2 weeks later violet proliferation dye (VPD)-labeled OT1 T cells were transferred. OT1 division was evaluated (Supplementary Fig. 3). **c** Quantification of the percent divided in the groups shown in **b**, where mice of each genotype were pooled from three independent experiments of three mice each. Statistical analysis was done using an unpaired $t$-test where the $p$-value between wild type and $BatF3^{-/-}$ is <0.0001 or wild type and $CCR7^{-/-}$ is <0.0001. Error bars are mean ± standard error of the mean

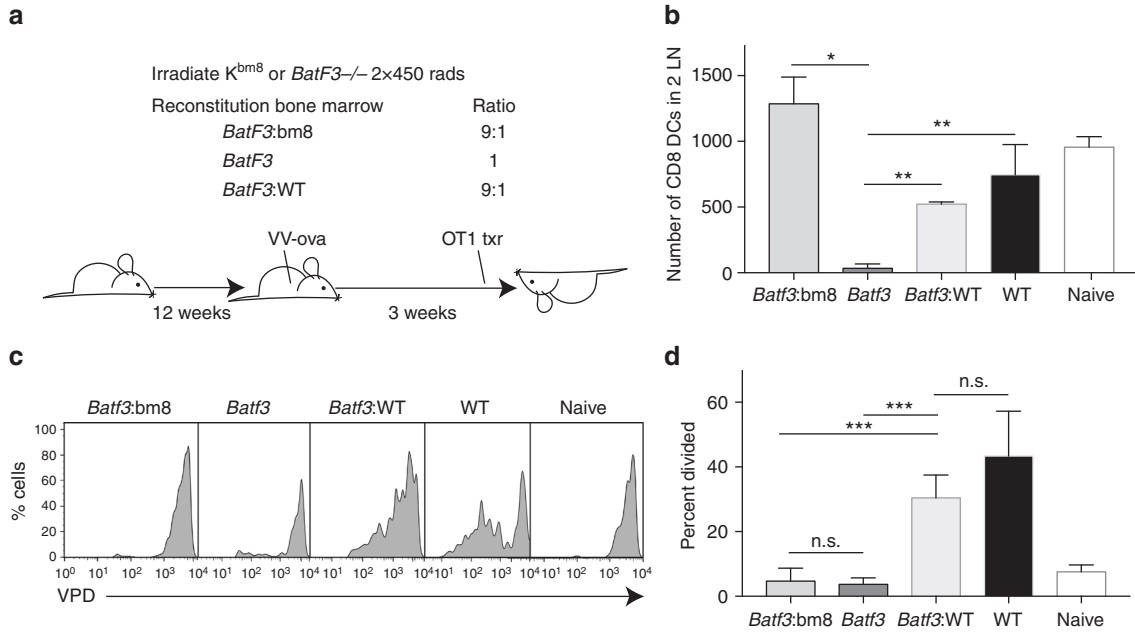

**Fig. 6** BatF3 DCs are necessary and sufficient for viral-archived antigen presentation. **a** Schematic of experimental design. Wild-type (WT) or $K^{bm8}$ mice were lethally irradiated and reconstituted with the indicated mixture of bone marrow. Following reconstitution, mice were infected with vaccinia expressing ovalbumin. Three weeks after immunization, violet proliferation dye (VPD)-labeled OT1 T cells were transferred into mice and T-cell division was assessed 3 days later. **b** Number of BatF3-dependent DCs was quantified using CD8 as a marker of BatF3 DCs in each strain used (Supplementary Fig. 3). Statistical analysis between groups was performed using an unpaired $t$-test, where BatF3: $K^{bm8}$ compared to BatF3 $p$-value was 0.0133, BatF3: $K^{bm8}$ compared to BatF3:WT was 0.0341, Batf3 compared to BatF3:WT was 0.0029, WT compared to BatF3 was 0.0059, and BatF3:WT compared to WT was 0.237. Shown are mean ± standard error of the mean. **c** Histograms of OT1 divided violet proliferation dye-labeled T cells from individual mice used in the chimera experiment. **d** Percent divided was calculated for all groups and statistical analysis between groups was performed using an unpaired $t$-test, where $BatF3$: $K^{bm8}$ compared to $BatF3$ $p$-value was 0.6125, $BatF3$: $K^{bm8}$ compared to $BatF3$:WT was <0.0001, $Batf3$ compared to $BatF3$:WT was <0.0001, and $BatF3$:WT compared to WT was 0.071. Shown are mean ± standard error of the mean. Six mice per group were evaluated from two separate lethally irradiated groups

facilitates LEC ova capture and archiving by LECs for at least 3–5 weeks after infection[24]. We infected $BatF3^{-/-}$, $Ccr7^{-/-}$, and WT control hosts with VV-ova and 3 weeks later assessed the exchange and presentation of LEC-archived antigens using VPD-labeled OT1 transfer (Fig. 5a). Similar to vaccination, antigen presentation was essentially ablated in the $Ccr7^{-/-}$ hosts, confirming the necessity of migratory APCs in the presentation of

antigens archived during virus challenge (Fig. 5b, c). Interestingly, antigen presentation was severely affected in the $BatF3^{-/-}$ hosts, with no detectable T-cell division at 3 weeks post vaccinia infection. These data demonstrate that the exchange of virus-derived antigens between LECs and APCs displays a greater reliance on migratory BatF3-dependent DCs after viral challenge.

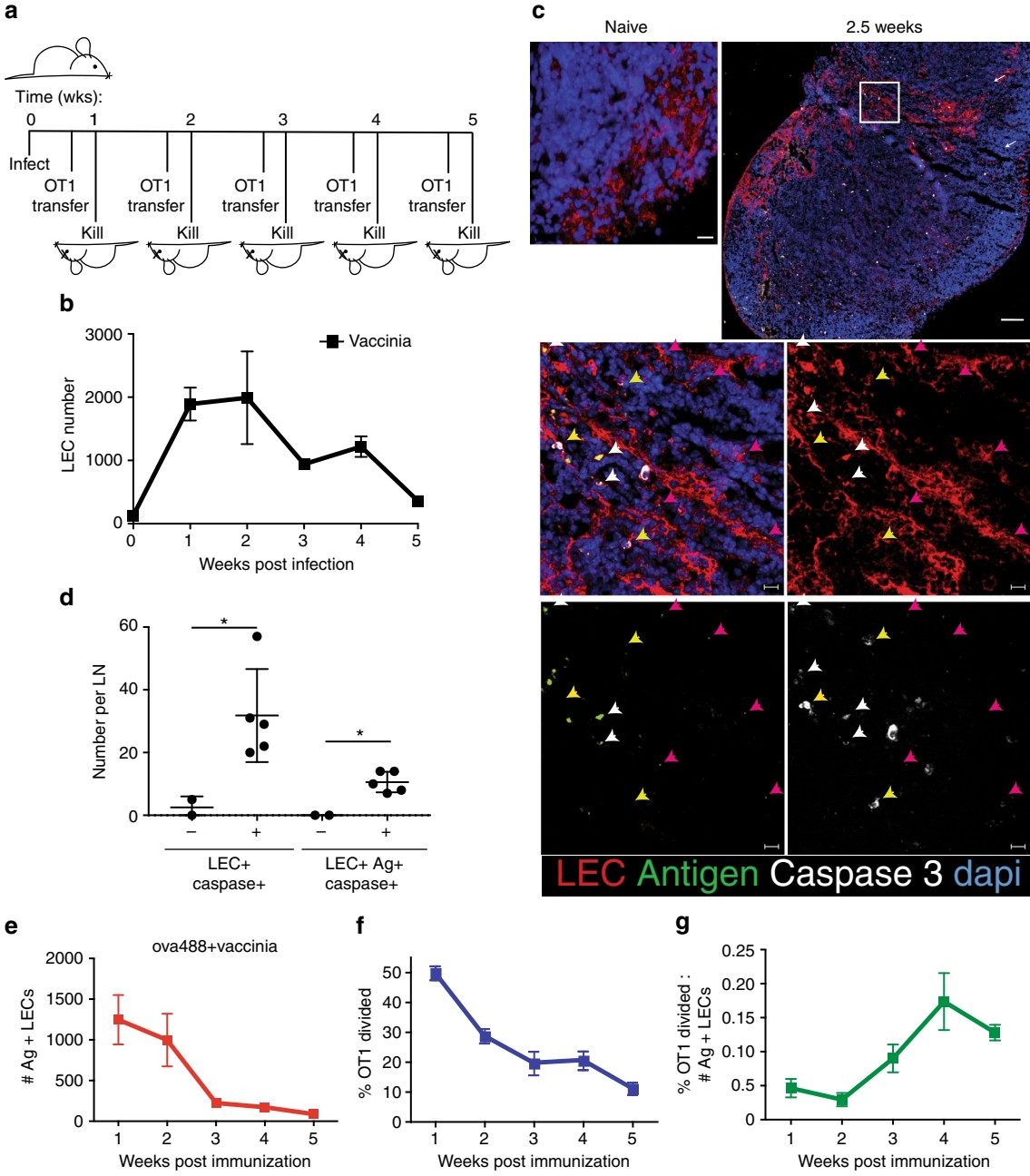

**Fig. 7** LEC apoptosis causes increased antigen presentation during LN contraction. **a** Schematic of experimental design for **b**, **e**–**g**. **b** Mice were infected weekly for 5 weeks with vaccinia virus ($1 \times 10^4$ colony forming units (CFU)) per foot and the number of LEC from the popliteal lymph nodes was determined to evaluate the increase in LECs over time. **c** Wild-type mice were immunized subcutaneously with ovalbumin conjugated to Alexafluor 488 (ova488) (10 μg) poly(I:C)/αCD40 (2 μg each) or not and killed 2.5 weeks later. Lymph nodes were removed and imbedded in optimal cutting temperature compound, sectioned, and stained with LEC marker Lyve-1 (red), cleaved caspase 3 (white), and dapi (blue), antigen-ova488 is visible by green fluorescence where indicated. Arrows in slide are as follows: white-antigen[+] caspase[+]lyve1[+], yellow-caspase[+]lyve1[+], pink-lyve-1[+]caspase[+]. Scale bar is 100 μm in top panels and 15 μm in zoomed panels. **d** Quantification of lymph nodes from **c**. Experiment was repeated thrice with five lymph nodes evaluated. An unpaired *t*-test was used to calculate the *p*-values for differences from naive which were $p = 0.046$ for LEC[+]Ag[+] and $p = 0.0076$ for LEC[+]Ag[+]Caspase[+]. **e** Mice infected weekly as in Fig. 1 with ova488 + vaccinia virus were killed and the percentage of antigen-positive LECs was calculated using flow cytometry. **f** OT1 T cells were transferred into mice infected weekly, as in **c**, 3 days prior to killing and the percent antigen-specific T cells divided was calculated. **g** The ratio of percent divided to number antigen positive was calculated from **e** and **f**. All experiments were performed at least twice with at least three mice per group per time point unless otherwise indicated. Error bars are mean ± standard error of the mean

We next devised a system to determine whether we could reconstitute viral-archived antigen presentation through WT reconstitution of the BatF3-dependent DC subsets. We again utilized a bone marrow chimera system similar to that shown in Fig. 1. In this case, the irradiated host was reconstituted with $BatF3^{-/-}$ bone marrow either alone or mixed in a 9:1 ratio with

WT or K[bm8] bone marrow (Fig. 6a). While >90% of all non-BatF3-dependent cellular subsets are derived from the $BatF3^{-/-}$ bone marrow, the entirety of the BatF3-dependent DC subsets is derived either from bone marrow that can (WT) or cannot (K[bm8]) present antigens to CD8[+] T cells. Though hosts reconstituted with only $BatF3^{-/-}$ bone marrow failed to

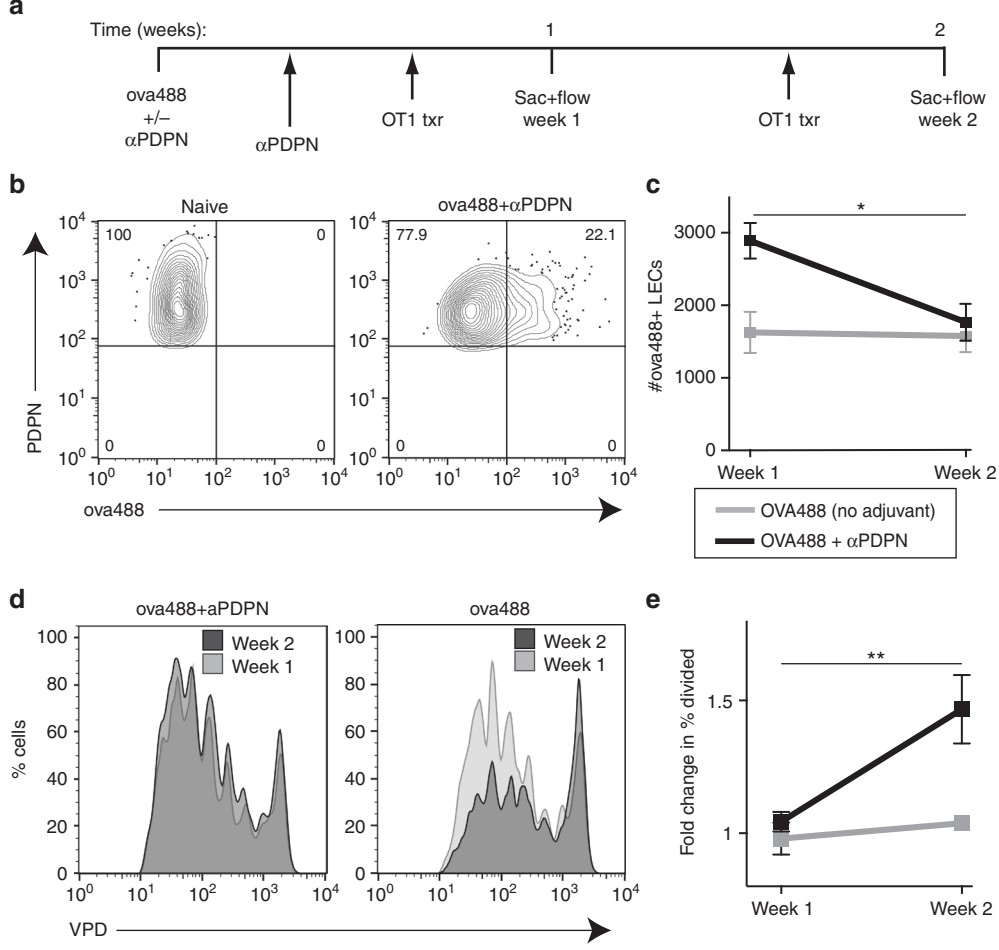

**Fig. 8** Loss of antigen-bearing LECs results in antigen presentation. **a** Experimental design using anti- PDPN to expand lymphatic endothelial cells. **b** Wild-type mice were immunized subcutaneously with fluorescent antigen (10 μg/site) along, or in combination with two doses of the anti-PDPN antibody (100 μg iv). One week after injection, mice were killed and the presence of antigen was visualized by flow cytometry. **c** Antigen⁺ lymphatic endothelial cells were quantified at 1 and 2 weeks after antigen and PDPN treatment as in **a**. Mice of each treatment were pooled from two independent experiments with three mice per group per time point, where error bars represent the mean ± standard error of the mean. Statistical analysis was done using two-way ANOVA with a $p$-value of 0.0123 represented using one asterisk. **d** OT1 T cells were transferred 3 days before killing at 1 and 2 weeks, and **e** their percent division was calculated and normalized to 1 in order to visualize differences between weeks rather than due to an effect on T-cell division caused by PDPN treatment. Mice of each treatment were pooled from two independent experiments with three mice per group per time point. Statistical analysis was done using two-way ANOVA with a $p$-value of 0.0031 represented using two asterisks in **e** T-cell division in antigen alone (gray shaded) and antigen with PDPN treatment (black line) 2 weeks post injection of antigen. Error bars shown in the figure are mean and standard error of the mean

reconstitute the CD8⁺ LN-resident or CD8⁺CD103⁺ migratory DC subsets, all mixed chimeras showed reconstitution of these lineages (Supplementary Fig. 6; Fig. 6b). We challenged these chimeras with VV and 3 weeks later quantified the antigen exchange by again monitoring presentation to transferred OT1 T cells. Similar to the intact $BatF3^{-/-}$ mouse, an irradiated host reconstituted with only $BatF3^{-/-}$ bone marrow failed to present archived virus-derived antigens (Fig. 6c, d). Similarly, antigen presentation was abolished when the BatF3-sensitive compartment was comprised of K$^{bm8}$ APCs (Fig. 6c, d). However, when WT bone marrow reconstituted the $BatF3^{-/-}$ compartment, antigen retention was restored to levels similar to WT (Fig. 6c, d). Combined with the data in the intact $BatF3^{-/-}$ host, we conclude that BatF3-dependent DCs are necessary and sufficient to present virally derived LEC-archived antigens.

**Viral-archived antigen presentation and LEC apoptosis**. Both BatF3-dependent and BatF3-independent migratory DCs in the lung can cross-present antigens to CD8⁺ T cells, though they

differ in what form of antigens they acquire[41,42]. While the BatF3-independent DCs can capture and present soluble antigens[42], the BatF3-dependent DCs are the only subset that can acquire apoptotic cell-associated antigens[41]. Assuming similar functions for these DC subsets in skin draining nodes, our data suggest that at least some of the vaccine-derived antigens are associated with the LEC in a way that is treated like soluble antigens from the perspective of the DC. Using the same logic, the critical importance of BatF3-dependent DCs for LEC-archived virus-derived antigens suggests that DCs may be acquiring antigens via uptake of apoptotic LECs. We sought to determine whether or not LECs underwent apoptosis during periods of antigen exchange between DCs and LECs. Groups of mice were challenged each week over the course of 5 weeks with VV in conjunction with Alexafluor 488-ovalbumin to track LEC-archived antigens (Fig. 7a). Three days prior to killing mice, VPD-labeled OT1 T cells were transferred into each group of immunized mice and 3 days later flow cytometry of the draining nodes was used to monitor the frequency and number of antigen⁺ and antigen⁻ LECs as well as the proliferation of the transferred

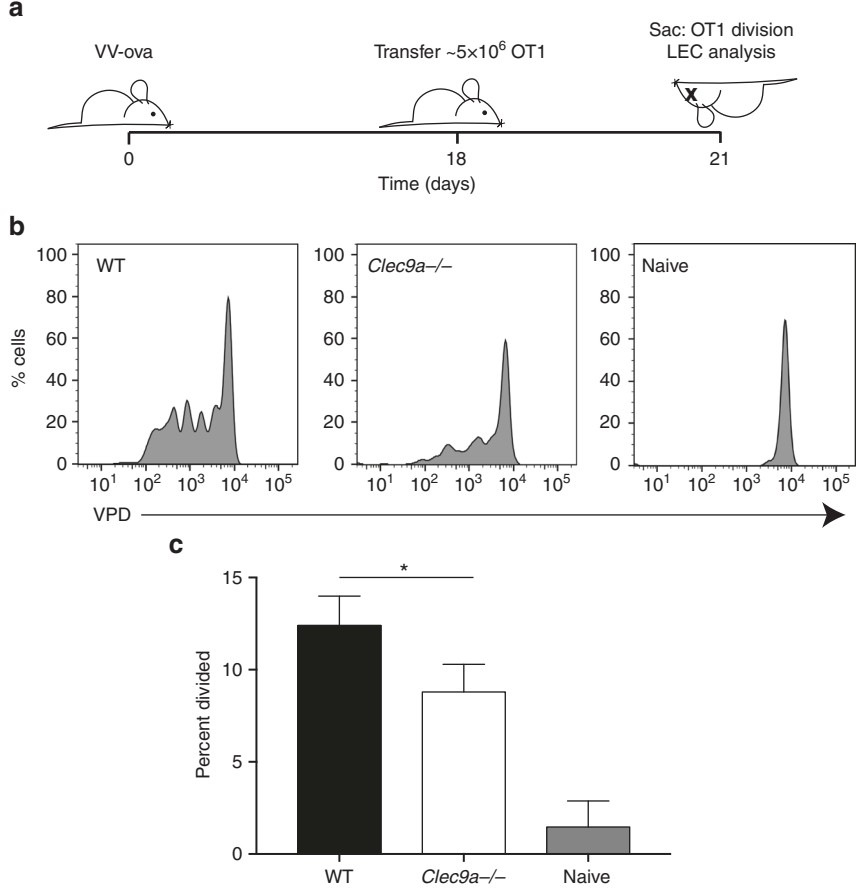

**Fig. 9** Antigen exchange following vaccinia infection utilizes Clec9a. **a** Schematic of experimental design. **b** WT and *Clec9a*$^{-/-}$ mice were infected with vaccinia-ova and 2 weeks later violet proliferation dye (VPD)-labeled OT1 T cells were transferred. OT1 division was evaluated. **c** Quantification of the percent divided in the groups shown in **b**, where mice of each genotype were pooled from two independent experiments of three mice each. Statistical analysis was done using an unpaired *t*-test where the *p*-value between wild-type and *Clec9a*$^{-/-}$ is 0.0305. Error bars are mean $\pm$ standard error of the mean

OT1s. Following viral challenge, there is a substantial increase (~10-fold) in the number of total LECs within the lymphoid tissue (Fig. 7b). We and others have shown that, over a range of immune challenges, roughly 30–40% of LECs undergo proliferation 3–5 days post immunization or infection, explaining the increase in LECs during this time frame[1–3,24]. Between weeks 2 and 5 after viral challenge, all LECs declined (Fig. 7b) and between week 2 and 3 following vaccination LECs became cleaved caspase 3$^+$ in both antigen$^+$ and antigen$^-$ LECs (Fig. 7c, d), consistent with LEC apoptosis occurring during LN contraction (Supplementary Fig. 7). Following infection, antigen-bearing LECs declined steadily over the course of the 5 weeks (Fig. 7e), during which the transferred OT1s always proliferated to some degree regardless of the time point of transfer but also declined steadily over time (Fig. 7f). However, when this was normalized to the number of antigen$^+$ LECs in each host, we noted a relative increase in proliferation at the time points in which antigen$^+$ LECs were in their greatest decline (Fig. 7g). As the proliferation of OT1s is a surrogate for LEC:APC antigen exchange, we concluded that the time points of greatest antigen-positive LECs apoptosis were also the points in which antigen exchange between LECs and APCs was the most efficient. Though indirect, these data in combination with our findings above suggest that migratory BatF3-dependent DCs would have access to apoptotic LEC-associated antigens derived from viral challenge for presentation to T cells.

We observed this correlation between LEC apoptosis and LEC:DC antigen exchange even under non-inflammatory conditions,

thus ruling out any involvement of adjuvant-induced or virus-induced inflammation. In vivo engagement of PDPN (using injection of a PDPN-specific monoclonal antibody) resulted in the relaxation of FRC contractility and subsequent FRC proliferation[4]. We used the PDPN antibody to examine the effect on LEC expansion, contraction, and concomitant-archived antigen exchange. Mice were injected with fluorescent antigen with and without two injections of anti-PDPN in the first week. VPD-labeled OT1 T cells were transferred 3 days before harvest which was either 1 week or 2 weeks after initial antigen injection, and both percent of antigen+ LECs and OT1 proliferation were determined by flow cytometry (Fig. 8a). As published for FRCs, antibody-mediated blockade of PDPN in vivo induced the proliferation of FRCs and LECs (Supplementary Fig. 8) and the subsequent acquisition of fluorescently labeled antigen by the LECs (Fig. 8b). During this period of LEC expansion and antigen acquisition (Fig. 8c), LEC:APC antigen exchange was no different than mice injected with antigen alone (Fig. 8d, e). However, as the PDPN-induced LEC numbers contracted back to control antigen-alone hosts (Fig. 8c), we observed a relative increase in proliferation of transferred OT1 cells (Fig. 8c–e). This is consistent with the conclusion that phases of LEC contraction are marked with increased antigen exchange and presentation to circulating CD8$^+$ T cells.

**Loss of *Clec9a* reduces archived viral antigen exchange**. We conclude that the capture and presentation of LEC-archived viral

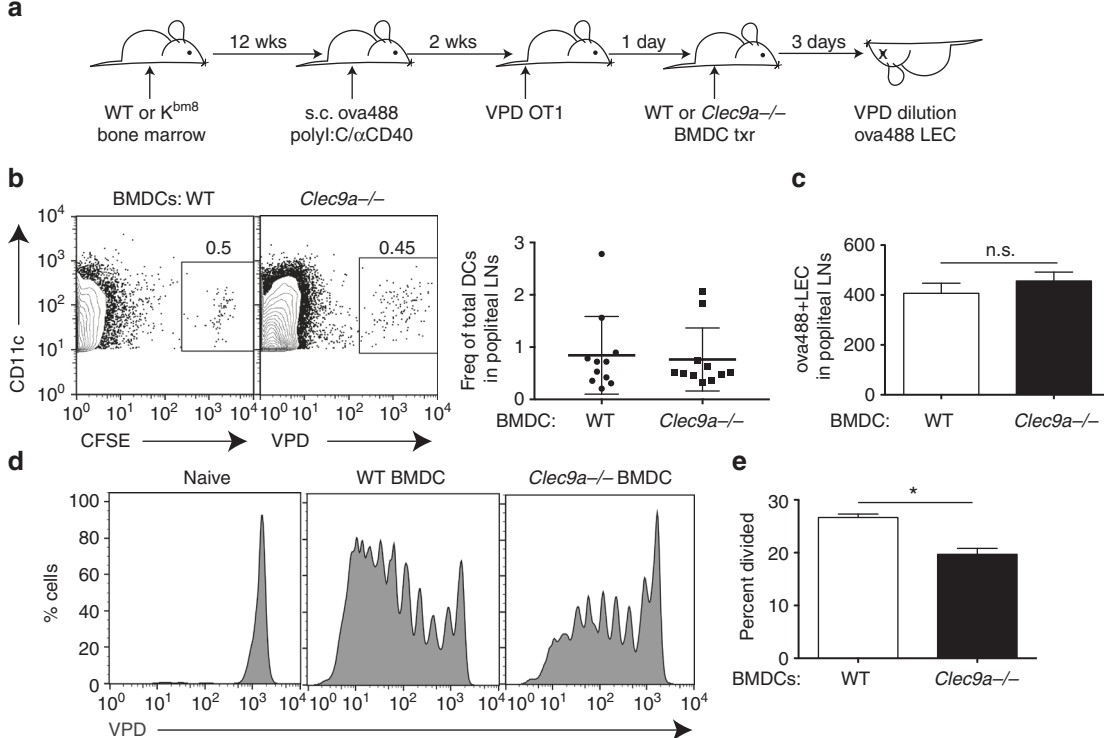

**Fig. 10** Antigen exchange from LECs to DCs utilizes Clec9a in a vaccine model. **a** Schematic of experimental design. **b** WT or *Clec9a*−/− BMDC were isolated from bone marrow and grown in vitro for 7 days. Wild-type BMDCs were labeled with CFSE and *Clec9a*−/− BMDCs were labeled with violet proliferation dye (VPD). BMDCs were transferred into the footpad of WT mice and frequency of CD11c+ transferred cells was quantified from each draining lymph node. Displayed are lymph node replicates and mean ± standard error of the mean. **c** Number of ovalbumin conjugated to Alexafluor 488 (ova488)-positive lymphatic endothelial cells was also calculated between groups with no significant difference by *t*-test. **d** As in Fig. 1d–f, mice were immunized subcutaneously in the footpad or not (naive) and wild-type or *Clec9a*−/− BMDCs were administered in the footpad 1 day after OT1 division and **e** percent divided was calculated. Shown are the chimeric mice from each genotype pooled from two independent experiments of three mice each, where the asterisk represents an unpaired *t*-test *p*-value of 0.009

antigens by migratory (CD103+), BatF3-dependent DCs occurs in the same time frame as LN contraction and LEC apoptosis. A growing body of evidence indicates that BatF3-dependent DCs, including CD103+ migratory DCs, are specialized for the acquisition of apoptotic cell debris[41,42,49]. These aspects of CD103+ DCs, along with our data above, suggested that acquisition of archived viral antigens may occur as a result of DC capture and uptake of apoptotic LECs. Though there are numerous receptors for the capture of apoptotic cells and debris, Clec9a/DNGR-1[41] is an efferocytic receptor which recognizes F-actin (only exposed during cell death[50]) and is highly expressed by CD103+ DCs. Importantly, *Clec9a*−/− mice have defects in cross-presentation of apoptotic cellular debris specifically in response to VV infection[51]. We therefore reasoned that loss of *Clec9a* on migratory APCs might strengthen the connection between apoptosis of LECs and archived antigen exchange and presentation in the context of viral infection. We challenged WT and *Clec9a*−/− mice with VV-ova in the footpad and 3 weeks later assessed the proliferation of transferred OT1s as an indicator of DC:LEC antigen exchange (Fig. 9a). Though antigen presentation was by no means ablated, the *Clec9a*−/− hosts demonstrated a reproducible and statistically significant ($p = 0.0305$, by unpaired *t*-test) reduction (~20–30%) in their presentation of LEC-associated antigens to the OT1 T cells (Fig. 9b, c). The magnitude and consistency of the Clec9a defect was somewhat surprising given that it is one of many apoptotic receptors on the surface of the cell. This decrease in antigen presentation was not due to decreased antigen archiving by the LECs, as similar numbers of antigen+ LECs were readily observable in knockout and control hosts (Supplementary Fig. 9d).

This defect in the *Clec9a*−/− host was not exclusive to viral antigen challenge, as a defect in *Clec9a*−/− mice was observed both following vaccination and even under non-inflammatory conditions when LEC expansion was induced via anti-PDPN treatment as in Fig. 8 (Supplementary Fig. 9). To evaluate the role of Clec9a intrinsic to the migratory DCs and in a model system where we can monitor LEC antigen archiving and DC migration, we again utilized the bone marrow chimera system described in Fig. 1, where the host DCs cannot present antigen to OT1 T cells. These mice were immunized with ovalbumin conjugated to alexfluor488 to provide fluorescent antigen for tracking archiving on LECs. Two weeks post immunization labeled OT1 T cells were transferred, followed by WT or *Clec9a*−/− BMDC transfer 1 day later (Fig. 10a). To first evaluate whether WT or *Clec9a*−/− BMDCs localized to the LN equally, WT or *Clec9a*−/− BMDCs were differentially labeled, injected into the footpad and their localization to the popliteal LN was evaluated (Fig. 10b). There was no difference in the capacity of the *Clec9a*−/− DCs to migrate to the node (Fig. 10b), nor was there any differences in the frequency of antigen-bearing LECs in WT or *Clec9a*−/− BMDC-injected hosts (Fig. 10c). However, transfer of labeled OT1 T cells again revealed a significant ($p = 0.009$, by unpaired *t*-test) defect in archived antigen presentation by the *Clec9a*−/− DCs compared to the WT (Fig. 10d, e). Collectively, these data indicate that efferocytic receptors such as Clec9a are utilized by migratory APCs to capture and present LEC-associated antigens. While this does not identify Clec9a as a mandatory receptor in the process of LEC:APC antigen exchange, in conjunction with the results above, it does directly link the

apoptosis of LECs to LEC-archived antigen exchange and presentation by APCs.

## Discussion

Collectively our data support a model whereby migratory DCs capture and cross-present antigens associated with LEC during LN contraction. These data provide mechanistic clarity to the processes by which LEC-archived antigens are managed within the lymphoid tissues for presentation to circulating memory CD8$^+$ T cells. Prior to the data presented here, it was formally possible that archived antigens, or portions of them, might either be released by the LEC and passively drained into T-cell regions of the lymphoid tissue, or brought there by migratory DCs/macrophages, to be picked up and presented by LN-resident DCs. The fact that migratory DCs can be directly shown to cross-present antigens exchanged with LECs, and that CCR7 and BatF3 are necessary for the presentation of archived antigens in vivo, argues that these migratory subsets serve to shuttle antigens from the draining lymphatics and into the T-cell regions of the node. This shuttling of antigens between LECs and DCs was previously demonstrated by Engelhard and colleagues as necessary for inducing CD4$^+$ T-cell tolerance to self antigens[20,24]. Our data now add to this original demonstration of DC:LEC antigen exchange, but with the outcome of enhancing protective CD8$^+$ T-cell immunity rather than enforcing tolerance[24]. We further provide evidence that DC uptake of apoptotic LEC is a contributing source of antigen for memory CD8$^+$ T cell maintenance.

Our present data provide direct evidence for the participation of both conventional DC1 (BatF3-dependent, XCR1$^+$) and cDC2 (BatF3-independent, XCR1$^-$) migratory DCs to both shuttle and present antigens following vaccination with antigens + adjuvant. Almost two decades ago, resident CD8$^+$ DCs were shown by Bevan and colleagues[52] to cross-present cell-associated antigens to CD8$^+$ T cells. The model system they utilized required some form of antigen exchange between the CD8$^+$ DCs and some other migratory cell[52]. More recently, Jakubzick and colleagues provided evidence that migratory DC subsets could capture and cross-present either soluble (BatF3-independent) or apoptotic cell-associated (BatF3-dependent) antigens in the lung[42]. Assuming these lung DCs are analogous to those found in the skin draining LN, our data support two conclusions in regard to how subunit vaccine antigens are archived by LECs; (1) at least some of the antigen associated with LECs is perceived by the cDC2s as "soluble"; (2) the remainder of the archived antigen is perceived by the DCs as "cell-associated". These two means of perceiving antigens by the DCs appear to be influenced by the immunological challenge experienced by the host. Unlike the bolus of soluble antigens delivered during subunit vaccination, viral antigens should be mostly associated with cells or cell debris, especially in the case of lytic infections such as VV. This difference correlates well with the DC subsets involved in antigen exchange with LECs; in contrast to subunit vaccination, the presentation of archived virus-associated antigens was ablated in both the BatF3$^{-/-}$ and the Ccr7$^{-/-}$ host. The fact that presentation of these antigens has a more restricted DC profile suggests that archived viral antigens are largely perceived as "cell associated" and are managed accordingly by only the BatF3-dependent DCs. In this regard, our data strongly implicate LEC apoptosis during LN contraction as the source of LEC:DC antigen exchange. Given that the draining LN expands ~8–10-fold more after viral infection than after subunit vaccination (Supplementary Fig. 7; Fig. 7), substantially more LEC death during LN resolution is expected. Consequently, a more robust restriction of viral antigen presentation to BatF3-dependent DCs might be more expected as well. That said, the presentation of subunit vaccine-related

antigens is always reduced in the BatF3$^{-/-}$, suggesting some role for LEC apoptosis in subunit vaccine antigen exchange as well. The influence of a single apoptotic cell receptor such as Clec9a on both vaccine (Fig. 10) and viral antigen (Fig. 9) presentation gives additional credence to a mechanism of antigen exchange centered on LEC apoptosis. Clec9a$^{-/-}$ DCs express a broad variety of other apoptotic cell receptors such as the class B scavenger receptor, CD36, that is important for phagocytosis of apoptotic cells[53]; T-cell immunoglobulin and mucin domain (TIM)1,3,4 which recognizes phosphatidyl serine[54] and CD205/DEC-205, which is involved in endocytosis and cross-presentation of apoptotic cell-derived antigens[55]. Given this range of apoptotic cell receptors, it is much more surprising that deficiency of just one of them resulted in a significant and reproducible reduction in vaccine and viral antigen presentation.

In using multi-photon to track DC:LEC interactions, we were surprised to find migrating DCs interacting for longer periods of time with antigen-bearing LECs than non-antigen bearing. Though our data here do not directly identify what might be responsible for this preference, the LECs that have captured antigens do have increased expression of adhesion receptors and chemokines that would logically increase DC migration and prolong LEC/DC interactions[56]. Our original publication showed the necessity for both inflammation and LEC expansion within the node in order for antigen capture and archiving by LECs to occur[24]. It seems likely that increased adhesive and chemotactic properties of the antigen-bearing LECs are a direct result of their response to inflammation and/or proliferation, and the possibilities are currently under investigation.

## Methods

**Mice**. About 6–8-week old female or male mice were purchased from either Charles river (C57BL/6) or Jackson laboratory (stock number 000664) unless otherwise stated and bred in housed in the University of Colorado Anschutz Medical Campus Animal Barrier Facility. WT (stock number 000664), K$^{bm8}$ [45,57,58], Clec9$^{-/-}$ (stock number 017696), Ccr7$^{-/-}$ (stock number 006621), proxtom (stock number 022766), actin-CFP (stock number 004218), and OT1 (stock number 003831) mice were all bred on a C57BL/6 background. BatF3$^{-/-}$ (stock number 013755) were maintained on a C57BL/6 (stock number 000664) X 129S1/Svlm (stock number 002448) cross that is roughly generation F2/3. OT1 mice are a TCR transgenic strain specific to the SIINFEKL peptide of ovalbumin (OVA257-264) in the context of H-2K$^b$. B6.C-H-2K$^{bm8}$ (K$^{bm8}$) H-2K$^b$ mutant cannot present the SIINFEKL peptide to OT1 T cells. K$^{bm8}$ mice were a kind gift from Larry Pease, Mayo Foundation, Rochester, MN. All animal procedures were approved by the Institutional Animal Care and Use Committee at the University of Colorado. Our calculations determined a group size of 5 can achieve a $p$-value of 0.008 when using the "permutation method"[59]. However, to minimize the number of animals, most experiments were performed with three mice per group if the $p$-value was <0.05 and repeated at least twice to bring the number of mice used to a number >5. Experiments were randomized for experimental vs. control groups.

**Vaccine and pathogen challenge**. Ovalbumin (Catalog number A5503, Sigma-Aldrich, St. Louis, MO) was decontaminated of lipopolysaccharide using a Triton X-114 detoxification method[60] and tested with Pierce LAL chromogenic endotoxin quantitation kit (catalog number 88282, Thermo Fisher Scientific, Waltham, MA). Ovalbumin was labeled with Alexafluor 488 using Alexafluor 488 succinimidyl ester labeling system (Catalog number A20100, Thermo Fisher Scientific, Waltham, MA). Ovalbumin was used in combination with poly(I:C) (catalog number VAC-PIC, Invivogen, San Diego, CA), and anti-CD40 (clone FGK4.5, BioXCell, West Lebanon, NH)[61,62]. For pathogen challenge, female C57BL/6 mice were injected with either 1 × 10$^7$ pfu/mouse of vaccinia virus western reserve strain expressing the ovalbumin protein (VV-ova) intraperitoneally or 1 × 10$^4$ pfu/footpad[24] using the vaccinia virus western reserve strain with ovalbumin labeled with Alexafluor 488. For anti-PDPN injection purified clone 8.1.1[4] (catalog number 127402, Biolegend, San Diego CA) was injected at 100 μg intravenously 4 h before ova-488 and 72 h later.

**OT1 transfer assays**. Mice were immunized with 0–10 μg of antigen (ovalbumin, Catalog number A5503, Sigma-Aldrich, St. Louis, MO or ovalbumin labeled with Alexafluor 488 catalog number A20100, Sigma-Aldrich, St. Louis, MO) with 1–2 μg poly(I:C) (catalog number VAC-PIC, Invivogen, San Diego, CA), and 1–2 μg anti-CD40 (clone FGK4.5, BioXCell, West Lebanon, NH)[61,62] per site into five sites

subcutaneously (footpads, flanks, scruff) or infected with VV-ova intraperitoneally $1 \times 10^7$ pfu/mouse or subcutaneously $1 \times 10^4$ pfu/footpad[24] 1–32 days prior to OT1 T-cell transfer. CD45.1 congenically marked OT1 T cells were isolated by the mouse CD8 isolation kit by Stem Cell Technologies (catalog number 19853, StemCell Technologies, Canada). Greater than 90% CD8 T-cell purity was obtained by this method. After isolation, OT1s were labeled with VPD (catalog number 562158, BD Biosciences) and $2 \times 10^5$–$1 \times 10^6$-labeled T cells were transferred into immunized C57BL/6 mice (CD45.2). Three days post transfer, mice were killed and spleens and draining LNs were harvested. Tissues were macerated with glass slides, red blood cells lysed with ACK (Ammonium–Chloride–Potassium) buffer, and stained with CD45.1 APC clone A20 (1:200), CD45.2 Alexafluor 488 clone 104 (1:200), B220 brilliant violet 510 clone RA3-6B2 (1:400), CD8 APC-Cy7 clone 53–6.7 (1:400), CD44 PerCP-Cy5.5 clone IM7 (1:400), and Vβ5.1,5.2 PE clone MR9-4 (1:300) to assess transferred OT1 T-cell proliferation (Supplementary Fig. 3). Determination of the cells divided (percent dividing cells) was calculated as previously described[63] using the equation fraction diluted $= \sum_1^i \frac{N_i}{2^i} / \sum_0^i \frac{N_i}{2^i}$ where $i$ is the generation number (0 is the undivided population), and $N_i$ is the number of events in generation $i$.

**Stromal and dendritic cell harvesting and staining.** Mice were killed 1–35 days post immunization or infection as above and popliteal, inguinal, axillary, and brachial LNs were harvested into EHAA media without L-glutamine (Gibco, Grand Island, NY). LNs were dissected using 22 guage needles to separate the tissue. Tissue was digested using 0.25 mg of Liberase DL (catalog number 5466202001, Roche, Indianapolis, IN) /ml of EHAA media and DNAse (catalog number LS002145, Worthington, Lakewood, NJ) for 1 h at 37 °C. An equal volume of 0.1 M EDTA in Hank's Buffered Saline Solution without calcium or magnesium (HBSS) was added and incubated for 5 min at 37°. Digested LNs were then pushed through a mesh screen and washed with 5 mM EDTA in EHAA. All flow cytometry antibodies were purchase from Biolegend (San Diego, CA) unless otherwise stated. Hematopoeitic cells were stained with B220 Brilliant Violet 510 clone RA3-6B2 (1:400), CD11c APC-Cy7 clone N418 (1:400), F4/80 APC clone BM8 (1:100), CD11b Pacific Blue clone M1/70 (1:400), and CD3 PE clone 17A2 (1:400). DCs were identified as CD11c$^{hi}$ B220$^-$, macrophages as CD11c$^-$ B220$^-$ CD11b$^+$ F4/80$^+$, B cells as CD3$^-$ B220$^+$, and T cells as B220$^-$ CD3$^+$ (Supplementary Fig. 1, top). Stromal cells were stained with CD45 Pacific Blue clone 30-F11 (1:300), Gp38/PDPN APC clone 8.1.1 (1:200), CD31 PerCp/Cy5.5 clone 390 (1:200). As published[64] stromal cell subsets were identified as follows: BECs, CD45– gp38– CD31+; LECs, CD45– gp38+ CD31+; FRCs, FDCs, and MRCs CD45– gp38+ CD31– (Supplementary Fig. 1, bottom). DCs were stained for CD11c APC-Cy7 clone N418 (1:400), B220 Brilliant Violet 510 clone RA3-6B2 (1:400), MHC class II APC clone M5/114.15.2 (1:600), CD11b Pacific Blue clone M1/70 (1:400), and CD103 PE clone 2E7 (1:100) to identify DC subsets (Supplementary Fig. 6; Fig. 4). Cells were run on the DakoCytomation CyAn ADP flow cytometer (Fort Collins, CO) and acquired using Summit acquisition software or the FACs Canto II and acquired using DIVA software (BD Biosciences, San Jose, CA). FlowJo software (Tree Star, Ashland, OR) was used to analyze flow cytometry data and cells were counted using a ViCell (Beckman Coulter, Brea, CA).

**T-DC coculture.** WT and BatF3$^{-/-}$ mice were immunized with 10 μg ova, 2 μg poly (I:C), 2 μg αCD40 in each of six sites—both footpads, both flanks, and on either side of the scruff of the neck. Two weeks later popliteal, inguinal, axillary, and brachial LNs were harvested and digested with Collagenase D at 2 mg/ml and DNAse (Worthington, Lakewood, NJ) for 30 min in EHAA media at 37° C. Digested LNs were pushed through a screen and washed with 5 mM EDTA in EHAA with 2% fetal bovine serum. Cells then underwent negative selection using anti-PE microbeads (catalog 130-048-801 Milltenyibiotech, Auburn, CA) and LS columns (catalog 130-042-401 Milltenyibiotech, Auburn, CA). Cells were stained with anti-PE NK1.1 (PK136), CD3 (17A2), B220 (RA3-6B2), and CD19 (6D5) to deplete NK, T, and B cells as directed by the Milltenyibiotech protocol. Following negative selection, cells were stained for CD11b PE-Cy7 (M1/70), CD11c BV421 (N418), MHC class II APC (M5/114.15.2), and B220 PE (RA3-6B2) and sorted on a BD FACsAria II cell sorter (BD biosciences, San Jose, CA) based on their CD11c and MHC class II expression (Fig. 4). Cells were then counted and equal numbers of cells (~20,000) were placed into different wells of a 96-well plate along with 50,000 VPD-labeled CD45.1+ OT1s isolated using a stemcell CD8 selection kit (catalog number 19853, StemCell Technologies, Canada) as described under OT1 transfer assays. Three days after co-culture, cells were stained with CD8 (53–6.7), CD45.1 (A-20), and CD44 (IM7) and analyzed on the Cyan ADP flow cytometer. Percent divided was calculated based on CD44 high expression.

**Bone marrow chimeras.** WT or BM8 mice were irradiated with 900–1000 Rads using a gamma irradiator with Cesium-137 as the source and rested for 4 h before reconstitution with either WT or BM8 bone marrow. Bone marrow was isolated and red blood cells were lysed prior to intravenous transfer. At 12 weeks post reconstitution, mice were immunized as described above. About 2–4 weeks post immunization and 1 day prior to OT1 transfer, WT or Clec9a$^{-/-}$ BMDCs were injected subcutaneously.

**BMDCs.** Bone marrow was isolated from the femurs of WT or Clec9a$^{-/-}$ mice. Cells were cultured in Modified Essential Medium (cellgro) with 10% FBS and 20 ng/ml of GM-CSF from the supernatant of B78hi-GM-CSF-expressing cell line (Dr. Hyam Levitsky, Johns Hopkins, Balitmore, MD)[65,66]. Concentration of GM-CSF was confirmed using an anti-GM-CSF ELISA capture antibody (purified anti-mouse GM-CSF clone MP122E9 at 100 μg/ml, eBioscience, San Diego, CA) followed by a biotinylated anti-GMCSF antibody. Concentration was determined using strepavidin-alkaline phosphatase (1:3000) and visualized at 405 nm on a Molecular Devices SpectraMax 250 (Sunnyvale, CA). Supernatant was filtered with a 0.2 μm filter prior to addition to media. Every 2 days, dead cells were spun down and half the volume of conditioned media was added to the cells along with half volume of 2× GM-CSF containing media. About 7–10 days later, BMDCs were harvested and counted and labeled with either CFSE or VPD. On average $1 \times 10^6$ BMDCs were transferred subcutaneously into each site.

**Fluorescence microscopy.** C57BL/6 or proxtom mice were immunized with 10 μg ova488, 2 μg poly(I:C), 2 μg αCD40 in the footpad and flank. Naive or mice immunized 2.5 weeks prior were killed and their popliteal or inguinal LNs placed in optimal cutting temperature compound or fixed in formalin and paraffin embedded. About 7–10 μm sections were cut using a cryostat or microtome and stained with an antibody against LYVE-1 (1:400 Abcam polyclonal) or LYVE1-APC (1:50 R&D systems clone 223322) and cleaved caspase 3 (1:100, Cell Signaling). Secondary antibodies used were anti-rabbit IgG-PE (1:200 Biolegend) and anti-rabbit IgG-dylight647 (1:200 Biolegend). Sections were imaged using a Nikon Eclipse Ti series fluorescent microscope at ×20 magnification with an aperture of 0.50. Images were taken with a Photometrics Coolsnap Dyano fluorescent camera. Images were analyzed using NIS elements AR software to evaluate co-localization. Multiple LN sections were analyzed from five LNs of infected mice or two LNs from naive mice and the number of cells co-localized was quantified per LN.

**Multi-photon microscopy.** Multi-photon imaging was performed using an Olympus FV1000MPE microscope with a XLPLN25XWMP Super 25 × 1.05-N.A. water immersion objective and a Spectra Physics 10-W Mai-Tai DeepSee-OL laser. Proxtom mice in which tdtomato was under the control of the Prox1 promoter were immunized with 10 μg ova488, 2 μg poly(I:C), 2 μg αCD40 12 days before killing. One day before killing, BMDCs from an actin-CFP mouse, grown as described, were injected subcutaneously into the footpad. Popliteal LNs were excised and fat was removed. Explanted LNs were immobilized to coverslips with efferent lymphatics adhered to coverslips. LNs were maintained at 35.5–37 °C in oxygenated RPMI. Excitation was performed at 910 nm. Actin-CFP BMDC fluorescence was captured by the 450–490 nm filter, ovalbumin conjugated to Alexafluor 488 was captured by the 500–550 nm filter, LEC tdtomato fluorescence was captured by the 575–640 nm filter for channel acquisition.

Images of 39–42 $xy$ planes (509 μm × 509 μm) with 3-μm $z$-spacing were acquired every min for 30 min. Image analysis was performed using Imaris (Bitplane) and Matlab (Mathworks) software. The channel arithmetics function in Imaris was used to perform linear unmixing between colors using images from single color control LNs acquired with the same instrument settings. Following linear unmixing, surfaces of LEC and BMDC populations were created and tracked using Imaris. Only BMDC surfaces tracked for ≥5 min were used for analysis. BMDCs were defined as being in contact with LECs if they were ≤2 μm from an LEC surface for at least two consecutive time points. Cumulative interaction times were determined by summing the total amount of time in contact with LECs for the length of the movie using Matlab. LECs that held antigen were identified by antigen spots that were completely within the LEC surface for the duration of the track.

**LEC sorting and qRT-PCR.** Five mice per group were immunized with 60 μg ova488, 30 μg αCD40, and 30 μg poly(I:C). About 12–14 days later, mice were killed and draining LNs were digested as described above. Cells were stained with CD45-PE. Cells then underwent negative selection using anti-PE microbeads and LS columns from Milltenyi and sorted on a BD FACsAria II cell sorter (BD Biosciences, San Jose, CA) based on the lack of CD45, PDPN+, CD31+, and fluorescent antigen (Supplementary Fig. 1, bottom). Antigen-positive and antigen-negative cells were run through a QIAshredder (catalog number 79656, Qiagen, Hilden, Germany) before RNA was isolated using an RNA micro kit (catalog number 74004, Qiagen, Hilden, Germany). Complimentary DNA (cDNA) was made using the Qiagen Quantitech Reverse Transcription kit (catalog number 205314, Qiagen, Hilden, Germany). Taqman Primers to CCL20 (Assay ID number Mm01268754_m1) and the control gene, GusB (Assay ID number Mm01197698_m1), were purchased from Thermo Fisher (Waltham, MA) and run on a Thermo Fisher Step 1 plus real-time PCR machine. cDNA was quantified using the delta-delta CT method to establish the fold increase in CCL20 from antigen + LECs compared to antigen-LECs. Experiments were completed three times and five mice per group were pooled for qRT-PCR. An unpaired t-test was used to determine statistical significance.

**Statistical analysis.** Statistical analysis was done using an unpaired Student's t test, two-way ANOVA, or linear regression analysis to determine the difference in

the slopes of two lines, in prism7 (GraphPad, San Diego, CA). *p*-values are denoted in the figure legend and in the figure images, where one asterisk represents a *p*-value of <0.05 and two asterisks a *p*-value of <0.01, and three asterisks a *p*-value of <0.001. Each analysis was done with at least three mice per group, per genotype, per time point, and each experiment was done at least twice with the same results and the data met the assumptions of the tests utilized for each experiment. For each experiment, there was no significant variation within groups and the variation between groups was similar. The investigator was blinded to the groups until analysis was complete to ensure results were unbiased. Error bars are mean ± the standard error of the mean.

**Data availability**. The data that support the findings of this study are available from the corresponding author upon request.

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

## Acknowledgements

We would like to thank Larry Pease for K^bm8 mice. This work was funded by NIH grants to BAT R01AI121209, RMK R01AI117918, and RSF JDRF 2-1012-197 and R01DK111733.

## Author contributions

R.M.K. contributed to the conception and design of experiments, interpretation of data, and drafting and revision of the article. R.S.L. contributed to acquisition of the data, analysis, and interpretation of the data and revision of the article, J.M.F. contributed to acquisition of the data, analysis and interpretation of the data, and revision of the article. E.D.L. contributed to acquisition and analysis of the data, and revision of the article, R.S.F. contributed to the conception and design of experiments and revision of the article. B.A.J.T. contributed to the conception and design of experiments, analysis and interpretation of the data, acquisition of the data, and drafting and revision of the article.

## Additional information

**Competing interests:** The authors declare no competing financial interests.

