## [Peer Review File · Nature Communications]

Reviewers' comments:

Reviewer #1 (Remarks to the Author):

This manuscript is a continuation of the previous work of the authors. In their earlier work they described antigen archiving on lymphatic endothelial cells (LEC) and the cross-presentation of antigen between LEC and some hematopoietic antigen presenting cell type. Now they have identified the cell type as migratory dendritic cells. In general the work seems to be well performed. The following points need to be addressed:

1. In several figures it looks that the results of a representative experiment have been presented. For example it is written in Figure 3: D. WT or BatF3^{-/-} mice were immunized 3 weeks prior to sacrifice and OT1 division was evaluated by VPD dilution. E. Quantitation of D. Representative experiment of 4 independent experiments with 3-5 mice per genotype. Statistical analysis was done using an unpaired t-test where $p=0.0016$. Error bars shown in the figure are mean and SEM

If my interpretation is correct, then the authors need to combine and show the results of all experiments besides the representative experiment. If my interpretation is wrong and in this case E is already a combination of the results of all experiments, please rephrase the wording. (This concerns also other figures).

2. It remains somewhat unclear, where the antigen cross-presentation is taking place – in afferent (roof or floor of the sub-capsular sinus) or efferent lymphatic arm (elsewhere in the node) or at both sites?

Minor:

-the language should be checked. There are several typos/mistakes such as (used to calculated; should be mostly associate with...)

-arrows in Figure 6C should be larger to increase their visibility

Reviewer #2 (Remarks to the Author):

In this study Kedl et al. have further investigated the mechanisms by which antigen (Ag) that is archived by lymphatic endothelial cells (LECs) in lymph nodes (LNs) during an immune response induces CD8 T cell proliferation at late time points (i.e. several weeks) after immunization. The authors had previously shown (Tamburini et al., Nat. Comm 2014) that T cell proliferation induced by Ag-archiving in LECs depended on CD11c⁺ antigen-presenting cells (APCs - experiments performed in CD11c-DTR BM chimeric mice). In the present work, the authors now show that Ag is transferred to migratory DCs (CCR7 dependence) and that this process is in part BaTF3-dependent. Moreover, the data presented attempt to establish a link between archived Ag transfer from LECs to DCs and LEC apoptosis, by demonstrating that Ag-induced T cell proliferation is reduced in clec9a-deficient mice and by correlating antigen-transfer with LN contraction (i.e. a phase of presumed massive LEC apoptosis). However, the experimental evidence provided for the latter hypothesis needs to be further solidified and expanded.

Major comments:

1) The point that migratory DCs take up antigen from apoptotic LN LECs is the most important novel claim of the study, but could benefit from some more experimental evidence. I believe this could be addressed by intravital microscopy (IVM), using the model already established for the experiments shown in Fig. 2: Using IVM the authors could investigate whether there are differences in the interaction of clec9a^{-/-} (or BaTF3^{-/-}) DCs vs. WT DCs with Ag-bearing LECs, what could support their hypothesis. Similarly, IVM could help to actually visualize antigen transfer from LECs to DCs. Somewhat puzzling for me, the authors show in Fig. 1C that at all time points, DCs are Ag-negative. At what time point do the authors then suggest that DCs could be Ag⁺ to allow for the stimulation of in vitro proliferation of OT-I cells?

2) The IVM data (Fig. 2D) suggest that migratory DCs in LNs preferentially interact with Ag⁺ LECs (ova-488⁺). This is not so straight-forward to understand, given that LN LECs will also uptake and archive other unlabeled endogenous antigen (e.g. proteins / macromolecules) present in lymph during the immunization. Thus, what induces DCs to preferentially interact with ova-488⁺ LECs? Could this observation be linked with the high dose of ova-488 antigen used in the experiments and perhaps be a sign of toxicity or linked with ova-488⁺ LECs going into apoptosis?

Moreover, regarding Fig. 2D, could the authors please clarify whether the increase in the cumulative DC-LEC interaction time is because individual cell-cell contacts are longer or whether DCs more frequently contact Ag⁺ LECs?

3) Fig. 6: The authors emphasize that T cell proliferation induced by archived Ag was increased at the time points at which Ag⁺ LECs were in their greatest decline (LN contraction). The evidence for this is not obvious when looking at Fig. 6F (% OT1 divided) but only comes from Fig. 6G, which however presents highly "processed" data, as the ratio of two percentages are shown (the ratio of %OT1 divided: % Ag⁺ LECs). Particularly in the case of LECs, which display such dramatic changes in total numbers over the course of the immune response (see Fig. 6B), the % of Ag⁺ LECs might be much less relevant than the absolute number of Ag⁺ LECs. The authors should therefore also show absolute numbers and analyze how the correlation (in analogy to Fig. 6G) develops in this case.

Referring to Fig. 6E and 6F the authors write: "Following infection, antigen-bearing LECs declined steadily over the course of the 5 weeks (Figure 6E), during which the transferred OT1s always showed some degree of proliferation regardless of the time point of transfer (Figure 6F)." => In the case of 6E, this statement can definitely only be made if absolute LEC numbers are shown (as requested above). In the case of 6E, it would be appropriate to also mention that proliferation was significantly declining over time.

4) The title and also the way results are presented and discussed suggest that Ag transfer from LECs to DCs and cross-presentation by DCs mainly take place during LN contraction. However, as already elaborated in 3) this is not supported by strong experimental data (only correlations shown in Fig. 6). In fact, OT-1 proliferation (Fig. 6F) seems to take place at an even higher rate in the early phases of LN remodeling (expansion, Fig. 6B). Thus, unless supported by more experimental evidence, the title and several statements in the manuscript need to be revised.

5) When referring to their previous findings in the abstract, the authors should disclose that they themselves have already identified the "previously unidentified hematopoietic-derived antigen presenting cells (APCs)" as CD11c⁺ dendritic cells (Tamburini et al., Nat. Comm. 2014). Moreover, in the spirit of a more balanced and transparent reporting and discussion of the novelty of their study, it would be appropriate if the findings of Rouhani et al., Nat. Comm 2015 and colleagues

were also mentioned and discussed; the latter authors have described a similar mechanism by which DCs take up endogenous Ag from LN LEC and present it to CD4 T cells in the context of tolerance induction.

6) The FACS histograms presented in Fig. 8B (WT and naive panel) appear to be identical to those presented in Fig. 5B (WT and naive panel). Can the authors offer a possible explanation for this? While it is plausible that the FACS plots duplicated in the manuscript come from the same experiment (i.e. that a comparison was made of WT, naive, BatF3-, CCR7-, Clec9-), it is unlikely that these plots are the most representable plots for both data sets, given that the average values of percent divided (or fraction diluted) shown in 5C and 8C are so strikingly different. - In any case, it is concerning if the FACS plots have been duplicated.

Minor comments:

1) Several statements in the text come without references of the corresponding literature. Examples are (but there are more):

- "Both Batf3-dependent and -independent migratory DCs in the lung can cross present antigens to CD8+ T cells, though they differ in what form of antigens they are able to acquire"

- "Importantly, Clec9-/- mice have defects in cross-presentation of apoptotic cellular debris specifically in response to vaccinia virus infection "

- "More recently, Jakubzick and colleagues provided evidence that migratory DC subsets could capture and cross present either soluble (BatF3-independent) or apoptotic cell-associated (BatF3-dependent) antigens in the lung."

- "Dendritic cell (DC) subsets can be reasonably split into three major groups; cDC1s, cDC2s and pDCs"

2) The manuscript requires better proof reading and editing. For example, in several cases there is a discrepancy between what is written in Figure legend and what is shown in the Figure (e.g. Fig. 5A indicates 18 days in the cartoon and refers to 2 weeks in the Fig. legend). Some graphs display a cropped y-axis (e.g. Figure 5C or Figure 6B). Moreover, abbreviations are often not introduced or used consistently.

3) The authors should provide the gating strategies for their FACS Figures, either directly in the Figure or in the Supplement. Moreover, please also write in the Figure Legend how the gating was performed. For example, from Fig. 1A,B it is not clear whether LECs were exclusively identified by podoplanin staining or - more likely - by gating on CD31+CD45- cells.

4) Fig. 1C: If possible, please provide absolute cell numbers rather than percentages, in analogy to Fig. 1B.

5) Fig. 4C: The Figure legend states that "the experiment was repeated twice with 6 mice per group". I do not understand how this correlates to the number of points in the graph. The next sentence states that "each point represents the number of wells evaluated" – Does this mean that

material from different 6 mice was pooled? It would probably be statistically more accurate in this case to investigate the same number of wells per each cell type.

6) Fig 4D: The authors comment in the text that BATF3^{-/-} mice had fewer total migratory DCs (CD11C1MHCII^{hi}) – but they are only showing % data in Fig. 4A: Please either provide total DC cell numbers or change the wording.

7) Fig. 5B: comparison of multiple groups cannot be analyzed by an unpaired t-test. Moreover, in the text it is written that “antigen presentation was if anything, even more severely affected in the BaTF3^{-/-} hosts”. Please provide the necessary statistical analysis for this statement in Fig. 5B.

8) Fig. 6A: The cartoon is somewhat misleading, since it gives the impression that there was only one group of mice that was repeatedly injected every week.

9) “Production of vascular endothelial growth factors by migrating mononuclear cells results in the growth of lymphatic vessels and blood vessels” To my knowledge, this is not the only mechanism, since also VEGF-A produced by B cells or drained from the inflamed tissue into the LN has been implicated in LN lymphangiogenesis.

10) “CCL19 and 21 made by FRCs can bind to the glycosaminoglycans in the afferent lymphatics,” To my knowledge, CCL19 only poorly binds GAGs (it lacks the positively charged C terminus). Moreover, CCL19/CCL21 produced by FRCs in LNs is unlikely to bind to GAGs in the upstream-located afferent lymphatic vessels (which produce CCL21 by themselves).

Reviewers' comments:

Reviewer #1 (Remarks to the Author):

This manuscript is a continuation of the previous work of the authors. In their earlier work they described antigen archiving on lymphatic endothelial cells (LEC) and the cross-presentation of antigen between LEC and some hematopoietic antigen presenting cell type. Now they have identified the cell type as migratory dendritic cells. In general the work seems to be well performed. The following points need to be addressed:

1. In several figures it looks that the results of a representative experiment have been presented. For example it is written in Figure 3: D. WT or BatF3^{-/-} mice were immunized 3 weeks prior to sacrifice and OT1 division was evaluated by VPD dilution. E. Quantitation of D. Representative experiment of 4 independent experiments with 3-5 mice per genotype. Statistical analysis was done using an unpaired t-test where $p=0.0016$. Error bars shown in the figure are mean and SEM

If my interpretation is correct, then the authors need to combine and show the results of all experiments besides the representative experiment. If my interpretation is wrong and in this case E is already a combination of the results of all experiments, please rephrase the wording. (This concerns also other figures).

I apologize for the wording here. The representative experiments are the flow files in A and in D while the quantification is a combination of results of all experiments. This has been changed in the figure legends to reflect this.

2. It remains somewhat unclear, where the antigen cross-presentation is taking place – in afferent (roof or floor of the sub-capsular sinus) or efferent lymphatic arm (elsewhere in the node) or at both sites?

We believe the cross-presentation occurs in the T cell zone of the lymph node where DCs encounter T cells. When we have imaged the lymph node and transferred in labeled T cells they interact with DCs in the center of the lymph node-which we assume is the T cell zone. However, the antigen hand-off likely occurs in the subcapsular sinus and antigen presentation may occur differentially between naïve and memory T cells. This is the work of future studies. We have added text to the discussion to clarify and a supplementary movie to show DCs interacting with antigen specific T cells imaged in the center of the lymph node of the same lymph node shown in figure 2.

Minor:

-the language should be checked. There are several typos/mistakes such as (used to calculated; should be mostly associate with...)

We have checked our manuscript for typos/mistakes and had it proofread by others.

-arrows in Figure 6C should be larger to increase their visibility

The arrows in figure 6C-now 7C have been increased in size.

Reviewer #2 (Remarks to the Author):

In this study Kedl et al. have further investigated the mechanisms by which antigen (Ag) that is archived by lymphatic endothelial cells (LECs) in lymph nodes (LNs) during an immune response induces CD8 T cell proliferation at late time points (i.e. several weeks) after immunization. The authors had previously shown (Tamburini et al., Nat. Comm 2014) that T cell proliferation induced by Ag-archiving in LECs depended on CD11c+ antigen-presenting cells (APCs - experiments performed in CD11c-DTR BM chimeric mice). In the present work, the authors now show that Ag is transferred to migratory DCs (CCR7 dependence) and that this process is in part BatF3-dependent. Moreover, the data presented attempt to establish a link between archived Ag transfer from LECs to DCs and LEC apoptosis, by demonstrating that Ag-induced T cell proliferation is reduced in *clec9a*-deficient mice and by correlating antigen-transfer with LN contraction (i.e. a phase of presumed massive LEC apoptosis). However, the experimental evidence provided for the latter hypothesis needs to be further solidified and expanded.

Major comments:

1) The point that migratory DCs take up antigen from apoptotic LN LECs is the most important novel claim of the study, but could benefit from some more experimental evidence. I believe this could be addressed by intravital microscopy (IVM), using the model already established for the experiments shown in Fig. 2: Using IVM the authors could investigate whether there are differences in the interaction of *clec9a*^{-/-} (or *BatF3*^{-/-}) DCs vs. WT DCs with Ag-bearing LECs, what could support their hypothesis. Similarly, IVM could help to actually visualize antigen transfer from LECs to DCs.

This point is well taken and we agree that the DC acquisition of apoptotic LECs is one of the more novel conclusions of our data. Our multiphoton experiments do visualize antigen transfer between migrating DCs and LECs! However, these are viable LECs, not LECs undergoing apoptosis. The quantitative difference that we see in multi photon microscopy is of DCs interacting for longer periods of time with LECs that retain antigen derived from subunit vaccination, not of DCs picking up dying LECs. As such, we would not expect same experiment with *Clec9a*^{-/-} APCs to be informative for LEC/DC interactions. As our data show, both *BatF3*-dependent and independent DCs can acquire antigen from LECs in the context of subunit vaccination. The multiphoton data obtained thus far with the BMDCs are most likely a model for LEC antigen acquisition by *BatF3*-independent migratory DC.

The experiment as proposed by the reviewer is actually problematic for a couple of reasons. First, as the LECs die, their membrane becomes permeable and they release their fluorescence such that we can no longer visualize them. Second, apoptotic cells are notoriously difficult to find in vivo, and we would need to establish a more robust model system to hope to capture these very rare and unpredictable events by multiphoton microscopy. The only real way to fully visualize DC acquisition of apoptotic LECs requires first validating a handful of available caspase 3 activated fluorescent probes for multiphoton. Assuming one works well for lymph node imaging, this would then have to be crossed to the *ProxTom* or *prox1-cre X stopflox-YFP*, followed by a cross to the *XCR1-TdTomato*. Last this host would then be left on WT or crossed again to the *Clec9*^{-/-}. This final host would then allow the visualization of endogenous *XCR1*⁺ (*BatF3*-dependent) migrating DCs as they interact with LECs that either are or are not undergoing apoptosis after viral challenge or vaccination. While this is ultimately a system we are presently pursuing, its ability to produce useful data once generated is still questionable, and it seems reasonable to suggest that its use lands well outside the boundaries of time and scope for the present manuscript.

However, in an effort to further bolster our conclusions, we have added additional data to the manuscript and supplementary figures that relate to the reviewers concern. Figure 10 shows the significant difference in T cell division between injection of WT or Clec9a^{-/-} BMDCs. New data in the figure now shows no defect in the frequency of BMDCs transferred between WT and Clec9a^{-/-} BMDCS, or in the frequency or number of antigen bearing LECs in the lymph node. Thus, the difference in T cell proliferation must be due to differences in the acquisition of the antigen, a difference that is attributable to the presence of the apoptotic receptor, Clec9a. We have also added a supplementary figure to show that there is a small but significant defect in subunit vaccinated mice that do not have Clec9a (supplementary figure 7A-D). Further in the second half of supplementary figure 7 (Supplementary figure 7E-H) we show that while the LECs in the Clec9a^{-/-} mice mimic the WT mice after PDPN antibody (Supplementary figure 7F) the increase in antigen presentation we see in the WT mice as the LECs die is absent in the Clec9a^{-/-} mice (Supplementary Figure 7H). These data confirm the data presented in the BMDC transfer experiment and in the intact Clec9a^{-/-} experiments in a system where inflammation is not present. These changes are highlighted in the text as well. Lastly, we have added data regarding the necessity of BatF3 in viral archived antigen exchange. In the new figure 6 and supplementary figure 4, we use a bone marrow chimera system in the context of viral infection, a setting where BatF3 sensitive DCs are more stringently required for archived antigen presentation than in vaccination. In these data, either BatF3^{-/-} or K^{bm8} hosts are reconstituted with bone marrow mixtures derived from WT, K^{bm8}, and BatF3^{-/-} mice. Reconstitution with BatF3 marrow alone or in a 9:1 ratio with to K^{bm8} or WT marrow produces hosts in which there are either no BatF3-sensitive DCs (BatF3:BatF3), or BatF3-sensitive DCs that can (BatF3:WT) or cannot (BatF3:K^{bm8}) present antigen to T cell (Figure 6B-shown as CD8 DCs). In this figure we show that the rescue of T cell division only occurs when the BatF3 dependent DCs are WT. These data indicate that WT BatF3-sensitive DCs are absolutely required for archived antigen presentation after viral challenge. In combination with the CCR7^{-/-} data these data indicate that reconstitution of the BatF3 dependent migratory DCs is necessary and sufficient for archived antigen presentation of vaccinia. As these DCs (CD103+) have a unique role in presenting apoptotic cell debris, these data further strengthen the point that migratory DCs could acquire antigen from apoptotic LECs.

Somewhat puzzling for me, the authors show in Fig. 1C that at all time points, DCs are Ag-negative. At what time point do the authors then suggest that DCs could be Ag+ to allow for the stimulation of in vitro proliferation of OT-I cells?

As the multiphoton data show, DCs likely become antigen + after interactions with antigen bearing LECs. The use of the large numbers of injected BMDCs in these experiments allows us to visualize this exchange, but the rarity of the event, along with the speed of antigen engulfment and processing, are likely reasons why this is difficult to visualize for endogenous DCs.

2) The IVM data (Fig. 2D) suggest that migratory DCs in LNs preferentially interact with Ag+ LECs (ova-488+). This is not so straight-forward to understand, given that LN LECs will also uptake and archive other unlabeled endogenous antigen (e.g. proteins / macromolecules) present in lymph during the immunization. Thus, what induces DCs to preferentially interact with ova-488+ LECs? Could this observation be linked with the high dose of ova-488 antigen used in the experiments and perhaps be a sign of toxicity or linked with ova-488+ LECs going into apoptosis?

This finding was also highly unexpected, and suggests that the LECs that have captured antigens either have increased adhesion receptors to prolong their interactions with DCs, or

produce chemotactic factors that increase DC migration and dwell time. Our original publication showed the necessity for both inflammation and LEC expansion within the node in order for antigen capture and archiving by LECs to occur. It seems likely that the LECs that have responded to this environment may possess one or both of the above mentioned properties, possibilities that are currently under our investigation.

In regards to toxicity, this does not seem to be the case for a few reasons. [Redacted]. Furthermore, we show in figure 7 that regardless of whether LECs retain antigen they become caspase 3 positive and we now show that total LEC numbers decline, not just antigen + LECs (figure 7B and E). Further, in the subunit vaccination model we have added LEC Ag+ numbers and LEC # to supplemental figure 5 to demonstrate that LEC Ag+ LECs don't preferentially die. Collectively, these all indicate that the antigen is not toxic.

Moreover, regarding Fig. 2D, could the authors please clarify whether the increase in the cumulative DC-LEC interaction time is because individual cell-cell contacts are longer or whether DCs more frequently contact Ag+ LECs?

The original figure was a measure of the total interaction time between DCs and LECs so we have added a panel to this figure to show the average length of each interaction time that a DC interacted with the LECs. See figure 2E. This data suggests that the DC-LEC interaction time is because cell-cell contacts are longer.

3) Fig. 6: The authors emphasize that T cell proliferation induced by archived Ag was increased at the time points at which Ag+ LECs were in their greatest decline (LN contraction). The evidence for this is not obvious when looking at Fig. 6F (% OT1 divided) but only comes from Fig. 6G, which however presents highly "processed" data, as the ratio of two percentages are shown (the ratio of %OT1 divided:% Ag+ LECs). Particularly in the case of LECs, which display such dramatic changes in total numbers over the course of the immune response (see Fig. 6B), the % of Ag+ LECs might be much less relevant than the absolute number of Ag+ LECs. The authors should therefore also show absolute numbers and analyze how the correlation (in analogy to Fig. 6G) develops in this case.

We thank the reviewer for this suggestion, and have included absolute numbers for total LECs (figure 7B), ova488+LECs (figure 7E), and the ratio of %OT1:# of Ag+ LECs (figure 7G). Indeed, given the steady loss of antigen-bearing LECs, one would initially expect that the OT1 proliferation should decline in a relatively analogous fashion. The fact that it actually rises when normalized to the ever decreasing antigen bearing LEC number strengthens our argument that LEC death is most likely a contributing factor to archived antigen presentation over time.

Referring to Fig. 6E and 6F the authors write: "Following infection, antigen-bearing LECs declined steadily over the course of the 5 weeks (Figure 6E), during which the transferred OT1s always showed some degree of proliferation regardless of the time point of transfer (Figure 6F)."

=>In the case of 6E, this statement can definitely only be made if absolute LEC numbers are shown (as requested above). In the case of 6E, I it would be appropriate to also mention that proliferation was significantly declining over time.

This has been added to the text and changed in the figure to add the numbers. Figure 6 is now figure 7.

4) The title and also the way results are presented and discussed suggest that Ag transfer from LECs to DCs and cross-presentation by DCs mainly take place during LN contraction. However, as already elaborated in 3) this is not supported by strong experimental data (only correlations shown in Fig. 6). In fact, OT-1 proliferation (Fig. 6F) seems to take place at an even higher rate in the early phases of LN remodeling (expansion, Fig. 6B). Thus, unless supported by more experimental evidence, the title and several statements in the manuscript need to be revised.

While a fair point is raised by the reviewer here, we believe the wording of the title is justified by the data, especially with the data modified as suggested by the reviewer. Contraction of the LN occurs between 2-5 weeks after vaccination or viral challenge, over which time, antigen presentation actually gains in efficiency relative to the number of antigen bearing LECs. Furthermore, the time points we examine in regards to the participation of BatF3-dependent and migratory DCs is within this time frame of LN contraction. Note we do not claim the exclusivity of antigen exchange and presentation during the contraction period.

5) When referring to their previous findings in the abstract, the authors should disclose that they themselves have already identified the “previously unidentified hematopoietic-derived antigen presenting cells (APCs)” as CD11c+ dendritic cells (Tamburini et al., Nat. Comm. 2014). Moreover, in the spirit of a more balanced and transparent reporting and discussion of the novelty of their study, it would be appropriate if the findings of Rouhani et al., Nat. Comm 2015 and colleagues were also mentioned and discussed; the latter authors have described a similar mechanism by which DCs take up endogenous Ag from LN LEC and present it to CD4 T cells in the context of tolerance induction.

Thank you for pointing out this oversight. The work of Rouhani et al is well regarded by our laboratories and was certainly in our minds as we wrote the discussion and manuscript and though we cited the Rouhani in the introduction it was an oversight by us not to include reference to this manuscript in the discussion. This has been corrected and text has been added to the introduction and discussion and has been bolded. Furthermore the abstract has been corrected to reflect that CD11c+ dendritic cells were previously shown to be required for archived antigen presentation.

6) The FACS histograms presented in Fig. 8B (WT and naïve panel) appear to be identical to those presented in Fig. 5B (WT and naïve panel). Can the authors offer a possible explanation for this? While it is plausible that the FACS plots duplicated in the manuscript come from the same experiment (i.e. that a comparison was made of WT, naïve, BatF3-, CCR7-, Clec9-), it is unlikely that these plots are the most representable plots for both data sets, given that the average values of percent divided (or fraction diluted) shown in 5C and 8C are so strikingly different. - In any case, it is concerning if the FACS plots have been duplicated.

We apologize for this inadvertent error in duplication and are extremely grateful that the review caught it! The manuscript went through many rounds of revision in regards to the order in which the data would be presented and what data was part of what figure set. The same graph inadvertently was duplicated and carried through the process without any of the authors (or other reviewers!) noticing. The correct version of each graph has now been properly placed in figures 5B and 9B.

Minor comments:

1) Several statements in the text come without references of the corresponding literature. Examples are (but there are more):

- “Both Batf3-dependent and –independent migratory DCs in the lung can cross present antigens to

CD8+ T cells, though they differ in what form of antigens they are able to acquire”

-“ Importantly, Clec9^{-/-} mice have defects in cross-presentation of apoptotic cellular debris specifically in response to vaccinia virus infection “

-“ More recently, Jakubzick and colleagues provided evidence that migratory DC subsets could capture and cross present either soluble (BatF3-independent) or apoptotic cell-associated (BatF3-dependent) antigens in the lung.”

- “Dendritic cell (DC) subsets can be reasonably split into three major groups; cDC1s, cDC2s and pDCs”

These references were cited elsewhere but not relative to the specific statements noted above; though they should have been. Again our thanks to the reviewer for bringing this to our attention. The citations have been placed in the text as requested.

2) The manuscript requires better proof reading and editing. For example, in several cases there is a discrepancy between what is written in Figure legend and what is shown in the Figure (e.g. Fig. 5A indicates 18 days in the cartoon and refers to 2 weeks in the Fig. legend). Some graphs display a cropped y-axis (e.g. Figure 5C or Figure 6B). Moreover, abbreviations are often not introduced or used consistently.

We have checked our manuscript for typos/mistakes and had it proofread by others.

3) The authors should provide the gating strategies for their FACS Figures, either directly in the Figure or in the Supplement. Moreover, please also write in the Figure Legend how the gating was performed. For example, from Fig. 1A,B it is not clear whether LECs were exclusively identified by podoplanin staining or - more likely - by gating on CD31+CD45⁻ cells.

Gating strategies have been added to the supplemental data for all facs plots shown (supplemental figures 1,3,4). How cells were identified has also been written in the figure legends and bolded.

4) Fig. 1C: If possible, please provide absolute cell numbers rather than percentages, in analogy to Fig. 1B.

Cell numbers rather than percentages have been displayed in figure 1C.

5) Fig. 4C: The Figure legend states that “the experiment was repeated twice with 6 mice per group”. I do not understand how this correlates to the number of points in the graph. The next sentence states that “each point represents the number of wells evaluated” – Does this mean that material from different 6 mice was pooled? It would probably be statistically more accurate in this case to investigate the same number of wells per each cell type.

This has been corrected in the figure and the legend and replicates have been combined in the quantifications in figure 4C.

6) Fig 4D: The authors comment in the text that BATF3^{-/-} mice had fewer total migratory DCs (CD11C1MHCII^{hi}) – but they are only showing % data in Fig. 4A: Please either provide total DC cell numbers or change the wording.

DC cell numbers have replaced frequency of CD11c+ cells (figure 4D).

7) Fig. 5B: comparison of multiple groups cannot be analyzed by an unpaired t-test. Moreover, in the text it is written that “antigen presentation was if anything, even more severely affected in the BatF3^{-/-} hosts”. Please provide the necessary statistical analysis for this statement in Fig. 5B.

Comparison was done between two groups, WT and BatF3^{-/-} OR WT and CCR7^{-/-} thus an unpaired t-test is appropriate. The statement that BatF3^{-/-} hosts was more severely affected has been removed as this was not statistically significant, but instead a trend and overstatement.

8) Fig. 6A: The cartoon is somewhat misleading, since it gives the impression that there was only one group of mice that was repeatedly injected every week.

The cartoon (now figure 7A) has been modified as requested.

9) “Production of vascular endothelial growth factors by migrating mononuclear cells results in the growth of lymphatic vessels and blood vessels” To my knowledge, this is not the only mechanism, since also VEGF-A produced by B cells or drained from the inflamed tissue into the LN has been implicated in LN lymphangiogenesis.

The text has been changed and cited to reflect that B cells also participate in LN lymphangiogenesis and was bolded in the text.

10) “CCL19 and 21 made by FRCs can bind to the glycosaminoglycans in the afferent lymphatics,” To my knowledge, CCL19 only poorly binds GAGs (it lacks the positively charged C terminus). Moreover, CCL19/CCL21 produced by FRCs in LNs is unlikely to bind to GAGs in the upstream-located afferent lymphatic vessels (which produce CCL21 by themselves).

These statements have been removed from the discussion.

Reviewers' comments:

Reviewer #1 (Remarks to the Author):

The authors have sufficiently addressed the concerns.

Reviewer #2 (Remarks to the Author):

The authors have nicely addressed most of my comments and concerns. However, a few points remain that in my opinion need to be addressed more thoroughly.

1) Related to Major Comment 1: The authors state in their response that they are able to visualize antigen transfer between migrating DCs and viable LECs. This is not clear from the images shown in Figure 2B or supplementary movie 1. Could the authors please provide more convincing images of this transfer? E.g. images first showing antigen (ova-488) in LECs and then orthogonal views demonstrating antigen (ova-488) inside a DC following antigen exchange; or a later point of the IVM movie where the antigen-carrying DC would be expected to move away from the LEC. Although these are viable LECs, these data would still greatly support the overall conclusion of direct antigen exchange occurring between DCs and LECs.

2) Related to Major Comment 2: The IVM data (Fig. 2D) suggests that migratory DCs in LNs preferentially interact with ova-488 + LECs. Although I can partly follow the authors' response, I still think that the result shown are puzzling and require further controls. In theory, both labeled (ova-488) and unlabeled antigen (endogenous or exogenous proteins) should be taken up and archived by LECs. Thus, what induces DCs to preferentially interact with ova-488+ LECs, as opposed to LECs that have taken up unlabeled antigen? Both types of antigen could be bound and internalized by LECs and both might trigger increased expression of adhesion receptors/chemokines. What induces this apparent specificity for ova-488 over unlabeled endogenous antigen? To fully answer this question and my underlying concern that this might be an experimental artifact, the authors should repeat these experiments using a different antigen, labeled with a different fluorophore.

3) Referring to Figure 7F (formerly Figure 6): The authors write in the rebuttal that they have now stated in the text that proliferation significantly declined over time. However, I cannot find this statement.

4) In several FACS histogram plots (e.g. Fig. 9B versus Fig. 8D), the authors switch between displaying # cell or % of Max on the y-axis. Please chose a consistent labeling. In general, the y-axis display of # cells ranges from 0 -15 events (e.g. Fig. 9B), indicating that extremely few events must have been recorded per condition. Can the authors please comment on this?

5) Concerning the legend for Figure 1: now that the authors have changed the data to absolute numbers (as opposed to percentage), this also needs to be changed in the figure legend(which still states percentage).

1) Related to Major Comment 1: The authors state in their response that they are able to visualize antigen transfer between migrating DCs and viable LECs. This is not clear from the images shown in Figure 2B or supplementary movie 1. Could the authors please provide more convincing images of this transfer? E.g. images first showing antigen (ova-488) in LECs and then orthogonal views demonstrating antigen (ova-488) inside a DC following antigen exchange; or a later point of the IVM movie where the antigen-carrying DC would be expected to move away from the LEC. Although these are viable LECs, these data would still greatly support the overall conclusion of direct antigen exchange occurring between DCs and LECs.

More convincing images and orthogonal views demonstrating antigen inside a DC following antigen exchange have been provided in Supplementary figure 3.

2) Related to Major Comment 2: The IVM data (Fig. 2D) suggests that migratory DCs in LNs preferentially interact with ova-488 + LECs. Although I can partly follow the authors' response, I still think that the result shown are puzzling and require further controls. In theory, both labeled (ova-488) and unlabeled antigen (endogenous or exogenous proteins) should be taken up and archived by LECs. Thus, what induces DCs to preferentially interact with ova-488+ LECs, as opposed to LECs that have taken up unlabeled antigen? Both types of antigen could be bound and internalized by LECs and both might trigger increased expression of adhesion receptors/chemokines. What induces this apparent specificity for ova-488 over unlabeled endogenous antigen? To fully answer this question and my underlying concern that this might be an experimental artifact, the authors should repeat these experiments using a different antigen, labeled with a different fluorophore.

The reviewer's question of course gets to the heart of one of the more interesting observations to emerge from the multi photon data; how/why do migratory DCs spend more time hanging around LECs that have taken up the antigen than around LECs that have not? It is worth noting that our data are not consistent with the interpretation that endogenous, unlabeled antigens must segregate into the fraction of LECs that are antigen negative. Just because the LECs have acquired exogenous antigens does not mean they do not also possess endogenous antigens, and in the absence of being able to track both, no claims can be made either way. Contained in the reviewers query are 2 concerns; i) is this migratory preference of the DCs specific to ovalbumin labeled with alexafluor488, and ii) assuming there is nothing strange about ovalbumin, what might be the real underlying reasons for this trafficking preference. While further multi photon experiments could be performed with alternate antigens, any conclusions made would only address concern (i). Instead, we have added data to the manuscript that we believe effectively addresses both (i) and (ii) above. First, while we showed that other antigens persist following immunization/infection in our last manuscript, we have now added supplementary figure 4 to show that a different antigen (BSA-HSVgB peptide (SSIEFARL) conjugate) with a different fluorofuor (alexafluor 647), produces results akin to what we see with Alexa-488 ovalbumin (antigen archiving, T cell division, etc). Second, we have discovered that ICAM1 and CCL20 are upregulated specifically within antigen bearing LECs, with the matching ligands LFA-1 and CCR6 being expressed by the BMDCs. Why antigen bearing LECs possess more of these molecules is under investigation, but may have something to do with the fact that LEC proliferation facilitates antigen capture (Tamburini et al. Nat Comm. 2014). We respectfully suggest that the further elucidation of this specific mechanism is out of scope for the present manuscript.

3) Referring to Figure 7F (formerly Figure 6): The authors write in the rebuttal that they have now stated in the text that proliferation significantly declined over time. However, I cannot find this statement.

This is highlighted within the manuscript in the description of figure 7F.

4) In several FACS histogram plots (e.g. Fig. 9B versus Fig. 8D), the authors switch between displaying # cell or % of Max on the y-axis. Please chose a consistent labeling. In general, the y-axis display of # cells ranges from 0 -15 events (e.g. Fig. 9B), indicating that extremely few events must have been recorded per condition. Can the authors please comment on this?

First, all of the plots have been changed to % cells. Second, the y-axis display is # of cells per each of 256 fluorescence intensity bins. So in our analysis it is not really the number of cells in each peak is as low as 15 cells as it depends on the number of bins the peak spans. The number of events collected for the proliferation analysis is typically between 200 and 1000.

See the link for more details: <http://docs.flowjo.com/vx/graphs-and-gating/data-visualization-and-display/gw-histograms/>

5) Concerning the legend for Figure 1: now that the authors have changed the data to absolute numbers (as opposed to percentage), this also needs to be changed in the figure legend (which still states percentage).

This has now been changed.

REVIEWERS' COMMENTS:

Reviewer #2 (Remarks to the Author):

The authors have adequately addressed my remaining concerns.